# 'College choice' under the COVID-19 pandemic: Sustainability of engineering campuses for future enrollments

Prashant Mahajan[1]*, Vaishali Patil[2]

1 R. C. Patel Institute of Technology, Shirpur, India, 2 Institute of Management Research and Development, RCPET's, Shirpur, India

* registrar.rcpit@gmail.com

**Data Availability Statement:** 1) All relevant data are within the manuscript. 2) It is deposited within data repository - figshare, with DOI 10.6084/m9.figshare.23521551.

## Abstract

Engineering profession for students and diverse students for Engineering Campuses (ECs) is the prestige to have for both. Worldwide higher education has been impacted by COVID-19 pandemic, but particularly pulling padlocked doors of Indian engineering campuses (ECs) down. Students' attitudes regarding choice, liking, and preferences were also affected. Knowing how tough 'college choice' was before the pandemic, one can guess how difficult it will be today. The objective of this study was to explore students' perceptions of choice characteristics related to ECs and diverse students enabling choice decisions under the COVID-19 situation, and to discover any possible relationships among them. Research questions were qualitatively examined with the statistical confirmation of related hypothesizes by utilizing ANOVA and Regression analysis. A self-reported quantitative survey composed of a closed-ended structured questionnaire was administered on the students of first-year engineering who had recently enrolled in ECs of North Maharashtra Region of India, after pandemic hitting India. According to the study, ECs have several characteristics impacting students' selection of ECs under pandemic. The influence of proximity, image and reputation, educational quality, and curriculum delivery was significant in contributing sustainability of ECs. This influence was significant across students' psychological and behavioural biases on likes, choices, and preferences. Furthermore, multiple relationships were noted within the sub-groups of demographic, geographic, socioeconomic, academic performance, and psychological and behavioural traits due to the impact of ECs' characteristics on sustainability. The study has provided a framework for policymakers and administrators to strengthen repositioning towards sustainability while capturing potentially diverse enrolments. Even if we have to coexist with pandemic forever or with more similar pandemics, the findings of this study may undergo a fundamental transformation for ECs (existing and forthcoming). On the other hand, by understanding the importance and relations of choice characteristics may smoothen the complex nature of "college choice" for prospective students.

**Funding:** The authors received no specific funding for this work.

**Competing interests:** The authors have declared that no competing interests exist.

**Abbreviations:** Abbreviation, Meaning; AICTE, All India Council for Technical Education, New Delhi; EC, Engineering campus; EE, Engineering education; HE, Higher education; HEI, Higher educational institution; IT, Information technology; ICT, Information and communication technologies; UGC, University grants commission, New Delhi; UNESCO, United Nations Educational, Scientific and Cultural Organization; WHO, World health organization.

## Introduction

By the time corona virus exploding COVID-19 pandemic began spreading throughout the globe in 2019–2020, every country had instituted a lockdown and outlawed economic exchanges involving direct human contacts. Because of the greater dependence of the education system on human connections, it too was forced to halt operations in an effort to shield its students and stakeholders from potential viral exposures, resulting significant impacts on Higher Educational Institutions (HEIs) in India and world [1,2] As a result of COVID-19, on short notice HEIs all across the world were prompted to perform 'online' which was the only concluding panacea. With the emergence of the COVID-19 pandemic in India, the Union Government proclaimed a nationwide lockdown that began on March 25, 2020. To comply with regulatory authorities' rules, Indian HEIs administrators were compelled to cease physical instructional activities and immediately switched to an online teaching-learning to complete the remainder of the academic year. Suddenly, the 'chalk and talk' style morphed into a 'click and collect' format. In India, the system of online education has never been tried at such a large scale and seemed to be like a massive social experiment. Regrettably, the 'online' approach had mostly cited as education in 'emergency' and not as the path of 'excellence'. It was no more an option but still was necessity [2]. Eventually, the situation due to COVID-19 in India in 2020 was such that education at all levels and disciplines were confronting the same storm but did not appear to be on the same boat. Educational leaders had somehow managed to overcome the difficulty by shifting to online. Nevertheless, by the time that they did so, they were faced with a millennium's supplementary challenge; a new enrollments of 2020–2021 academic year and dilemma of how these aspirants reimagine 'college choice'.

As acknowledged in earlier calls during non-pandemic situation, 'college choice' has been a difficult and complex [3–5], and mysterious process [6] encompassing a diverse human capital [7], social capital [8], and a broad and in-depth encounter of choice characteristics [9]. Prospective students live and grow for their life-long dream of 'college going', which does not happen in a vacuum for them. It is linked to a diverse set of traits escorted by students and colleges [10,11] it impacts on everyone; students, family members, policymakers of HEIs [12]. Students look into a wide range of options where they can devote their time, money, and dynamism with the utmost likelihood of experiencing satisfaction and success. While, HEIs look for students who will add to their campus's vibrancy and intellectual depth and who have a diverse interest both inside and outside the classroom. As a result, 'college choice' decisions are vital as it has a long lasting effect and can change students' life and HEIs' functioning forever [13].

Research on engineering education (EE) had has a little attention and is practically lacking on the concept of college choice, since the study trend appeared to be geared towards other higher education (HE) disciplines [14]. Before the pandemic hit, most engineering campuses (ECs) in India with low enrollments were in the poorest possible condition due to the lack of low enrollments and insufficient assessment of students' intentions. Until date, EE has been marked by students' multifaceted; growing and changing attitudes regarding staff and teaching quality, technological, programme value and delivery, outcome benefits, information and influence impact. Reports on EE available before the pandemic in regard to dwindling enrollments [15] in India as well as declining interest and trends around the world [16,17] have emphasized concerns for the overall development of techno-society. Due to non-availability of fresh enrollments and meager performance of ECs, the All India Council for Technical Education, New Delhi (AICTE), the apex regulatory body controlling technical education in India, was forced to close 6% of ECs and reduce intake capacity by 11.5%, over the two years to earlier to pandemic. Yet, the Indian ECs were unsuccessful to fill-up their intake capacity fully, leaving over 50% of seats allocated for new enrollments unfilled before the pandemic [18].

Engineering and technology is reported to be the fourth largest programme (previously the third largest) in 2020–2021, contributing 11.9% of total HE programmes in India, with the proportion of deprived population; female, minority, and lower socioeconomic status (SES) dropping compared to the pre-pandemic year 2019–2020 [19]. Despite being the world's leading producer of engineers, India still has a lower engineers-to-population ratio when compared to other developing countries [20], because of wide-ranging challenges such as awareness, diversity inclusion, and service quality [21–23] Although there is a significant demand for engineers, just a few of Gen alpha students are contemplating a career in engineering as a major. One can only speculate as to how difficult it will be for ECs to recruit students at this period, knowing how tough it was before the pandemic.

As the pandemic felt to be overextended over 2020, alarms were being raised world-wide on declination of new enrolments. Reports of McKinsey and Company [24], National Student Clearinghouse [25] and The International Association of Universities [26] indicated new enrolments estimates for academic year 2020–2021 were dire, predicting a drop of a double-digit percentage over forthcoming years, just making to block the entry of aspiring students of first generation and for four-year colleges. Most of the HEIs financial health started to touch bottom line due to not receiving accurate number of new students. The experts' concern about persistent declines was that the changing attitudes towards HE, which might jeopardize a generation's economic trajectory. History recalls that HE required fundamental restructuring in response to natural calamities [27] and pandemics [28]. Furthermore, these HEIs appeared to be struggling to get back on track and were highly influenced by situational crises including political, economic and environmental changes [29].

It is generally agreed that a career in engineering has the potential to have a long-term benefits on individuals' abilities [30], earning potential [31], social and the quality of life [32], industrial growth and country's development affecting national economy [33,34]. This is the reality that new generation born into a technologically advanced period will accomplish feats that earlier generations could only dream of. Ingenuity on the part of engineers has made this possible, which is great news for the general public. EE as a foundation for innovation, is essential for civilization's evolution due to the various risks that societies throughout the world face and that has a potential of the long-term and economic development of civilization, particularly in emerging nations like India. Diversity within EE is substantially more than only a "prestige to have". From its inception, EE has been a bastion of pedagogies that foster analytical thinking [35], which contributes in several ways to technological innovation and development via the use of a wide range of skills and perspectives of diversity. Nevertheless, it is found that embracing diversity with shifting interests is a challenge itself.

Coming to the current pandemic scenario, the ideas of 'one-size-fits-all' is definitely not be relevant in a country as diverse as India, with several educational dilemmas to tackle in the COVID-19 situation. Because there is so much at stake, there is a necessity to pivot in order to refocus and reposition that attracts the Gen alpha youngsters to the engineering path. Several studies have counseled the need of consistent assessment of college choice to sustain changing pattern of both; students as well as HEIs arising due to economic, environment, industrial and social needs. Accordingly, an evaluation of students' intent during the COVID-19 pandemic is critical and informative for legislators and EC administrators.

As informed by the evidence discussed above, the primary objective of this study to investigate perception of students on choice characteristics connected to diverse students and ECs facilitating college choice decisions under the COVID-19 pandemic, and discovering possible relationship between them. The research objective is supported by the three research questions pertaining to the selection of ECs during the COVID-19 pandemic.

*RQ1*. What are the perceived ECs characteristics essential for enabling EC choice under the COVID-19 pandemic? How it differs across students' diversity?

*RQ2*. How sustainability is related to the characteristics of ECs?

*RQ3*. What is the association in between sustainability and characteristics ECs within and across diverse students?

## Literature review

In order to fully investigate the history and development of the college choice phenomena, this study searches, selects, and integrates literature using a transparent, systematic, and comprehensive way. According to several theoretical and conceptual frameworks, college choice is complex [36] and has changed through time [37]. The phenomenon reaches a climax when student attributes equals college features, i.e., when students expectations meet HEIs proposals [38], resulting in a collective decision about 'college choice' [39]. It depends on students' knowledge of college facilities, their abilities and perceptions, and the help they receive from those around. [40]. Hence, 'college choice' depends on student and HEI choice characteristics, as illustrated below.

### Choice characteristics

Students' characteristics assist HEIs to select culturally compatible students and HEIs characteristics contribute to meeting students' demands. So, both types of characteristics must be recognized as they occur concurrently and analogously in the "college choice" decision-making process. Many and effective research on HE fields [11,36,41] have offered a complete picture of "college choice," justifying proper emphasis on myriad choice characteristics, while engineering studies have paid less attention. To create a route for potential students, HEIs must first understand students, what they expect from them, and how learning opportunities could fulfil those expectations [41]. The next section covers in depth what is known about choice characteristics related to HEIs and students, accounting for college choice.

### Students' characteristics

Though HEIs cannot control students' personal traits, which constitute selection triggers, but can understand it [42]. Hence, student prognosis is needed to assess HEI readiness for enrollment. Demographics and other factors influence college choices, according to several studies. That list includes students' demographic characteristics [43], such as gender; socioeconomic SES characteristics [44], such as social class; geographic characteristics [45], such as location; academic characteristics [46], such as academic performance; and psychographic and behavioral characteristics [47], such as interest in majors, course delivery priority, information preferences, and human influence.

### Demographics

Students' demographics are measurable. Gender is the most studied, researched, and useful attribute [48–50] and a significant predictive factor [51] in college choice in the benefits of HEIs HEIs. Gender differences in HE are evaluated by benefits/outcomes [52], finance [53], human influence and support [54,55], cultural aspects [56], safety and security [57], physical and social atmosphere [50,58], socioeconomic context [59], information sources [60], and behavioral and psychometric settings [61]. Gender equality in college choice is essential for sustainable economies [62].

College choice is a gendered process affecting their STEM sector entry [63–65]. With gender differences in college choice have several grounds like masculine, engineering was predominantly male domain [59]. Females remained underrepresented worldwide from EE due to feminine traits with lower self-efficacy [66]. Environmental variables has always affected college quality by gender [50]. Lower self-confidence [67], direct discrimination, cultural constructs, and identities [68] restricted female from STEM fields. Human motivation and support [56], particularly from family [69], has been a primary enticing tool for female students. Programme perks associated with as computer related engineering [70] urge male students to pursue EE [61]. Gender gaps are narrowing in India's HE, progressive a sign on India's equity policies [71], but this is not the case for engineering schools [72]. They make up around half of any country's population, but their involvement in engineering is substantially lower [73,74]. Female enrollment in undergraduate engineering programmes in India is presently approximately 30% [15] making them most underrepresented demographic in the engineering field.

## Socioeconomic status (SES)

Socioeconomic status (SES) refers to a combination of cultural, economic, and societal factors [46,75] and is the second most important constraint in college choice decisions [76–78] Compared to students from higher SES, students from lower SES are less likely to pursue STEM education [79]. Disparities on SES is wider in professional programmes like engineering, mostly affecting students from low-income rural areas, and females [80].

Higher SES families are more likely to invest in their children's education and encourage them to pursue a career in engineering, as reported in [56,81] The study of [79] found that families from lower SES need more access to career guidance and counselling services to decide which EC to enroll in. Several studies revealed that students from economically, socially, and culturally marginalized backgrounds have uneven and ecologically limited access to ECs. Undergraduate enrollments in engineering in India are a reflection of this transition, with only 17% of eligible students enrolling in engineering programmes for the 2019–2020 academic year [15].

## Geographic

Geographical factor the most important element in college choice decision [82], vary by location and is ruled by individual attitudes and beliefs, the availability of infrastructure, access to information in the area where the college is located. In India, urban and rural inhabitants shape the country's geography [83]. Rural populations have a lower SES and live in low-density villages, whereas urban populations have a diverse social background and live in densely populated towns and cities, but they have better access to basic human needs, cultural possibilities, and a larger economy [84].

Due to geographical isolation, loss of community communication [85], depression stigma [86] and English language anxiety [87], rural communities are more prone to deny HE. As a result, rural families are underserved in terms of knowledge and understanding concerning EE [88]. Rural students rely on self-motivation [89] and social support [85] from colleagues, and HEIs staff to support their college decision drive. Rural Indians make up 65.53% of the entire Indian population, with around 30% of rural students pursuing engineering career [15].

## Pre-college academic performance

The studies of [90–93] predicted that pre-college i.e. high school performance is important for pursuing engineering studies. High performance, particularly in science and mathematical

subjects, is the factor responsible for the establishment of STEM education interest at high school level [94,95]. Students must succeed in mathematics and science topics in order to be admitted to demanding majors or prestigious ECs in India. So, it becomes fundamental importance and concern for EE, where male students are biologically bred to outperform female students [96] in science subjects.

### Psychological and behavioral traits

These traits reflect students' intellectual and behavioral perspectives on their attitudes, personalities, values, learning perceptions, motivation, lifestyles, and preferred communication styles, all of which are reactive to college choice decisions [97]. The term educational motive describes a student's underlying reasons for engaging in particular psychological and behavioral action, including both their intrinsic character (student behavior driven by internal stimuli) and their extrinsic character (student behavior driven by external stimuli) [98]. Gaining insights into the key dimensions of students' behavior pattern is important for administrators and policy formulators as HEIs has become market driven [99], detailed as below.

### Attitude towards curriculum delivery

According to studies [100–103] students anticipate curriculum delivery to improve their access to technology-based methods of learning, rather than being antiquated and restricted by available resources and limited human interaction. Courses in education can be delivered in a number of different methods, including entirely online (with no in-person contact) [104], entirely on campus (with in-person contact) [105], and in a hybrid style [106,107] combining aspects of both. The relative importance of these options relies on factors including budget, timeline, desired results, and the nature of the offered services related to programmes. However, a program's curriculum delivery will be favored when it develops students' multi-disciplinary abilities to meet the demands of diverse students [108] and to sustain its value [109]. Deliberate curriculum delivery consists of intangible services and physical facilities that guarantees and monitors subject-matter expertise, transforms knowledge and abilities to fit a diversity of student needs [110].

### Liking for engineering majors

Historically, programme major choices have been influenced by technological and economic crises [11,111,112] Students' perceptions of certain jobs, as well as their ideas about their talents to provide an acceptable match, are critical in choosing an academic major [113]. Prospective students choose degrees that will benefit them in terms of career opportunities, employment returns, and academic achievement [114–116] In general, societal pressure, employment opportunities, exposure to and apply new technology, and a desire to learn all impact engineering major choices [117,118]. The prestige, better pay, and variety of career opportunities available to computer allied majors are well-known around the world [119], and as a result, prospective students are more willing to accept ECs with computer-related majors. Gender norms, feminism, and muscularity all have a role in engineering major selection [120]. Core majors like mechanical, chemical, civil, and electrical engineering were historically popular in India, mostly dominated by male. With the development of telecommunications and information technology, majors in electronics, computers, and related subjects have grown in popularity, particularly among females. More advanced and emerging majors, associated with high risks but great rewards, tend to attract students belonging to higher SES, good academic abilities, a high level of cultural awareness and technical proficiency, and urban regions [121].

Students from underprivileged backgrounds, those from rural locations, and those who struggled academically in high schools tend to stick with more traditional majors [122].

## Perceptions towards human influence

The effectiveness of the HE system relies heavily on human services as performances and humans as performers for the motivational impact they transmit with others. Numerous studies, for example [123–125] investigated the role that peoples' influence play in EE. Primary and long-lasting influences on a student's college choices include members of the student's immediate and extended family [56,126–129], teachers and counsellors from the student's high school [92,130], and supportive peers [131–133] Human impact considerably strengthens the "moment of truth" [134] and "word of mouth" [135]. The process of selecting an EC, which begins in high school [136], is one that is more successful and well-suited for women. It's true that certain high schools are better than others at inspiring their students to choose a career in engineering, but not all of them [137]. By engaging in two-way communication [52] about the location, safety and security, academic standards, and teaching quality of ECs [135,138], existing enrolled students, who are the real deal when it comes to teamwork and experience, can inspire aspiring students with a similar cultural and social background. Being human products and witnesses of their alma mater, alumni are the best sources of data about a HEI's notoriety, student life, and career prospects [139]. Knowledgeable parents are more likely to reach out to alumni and current students for details on housing options and safety measures for their female kid, as well as the academic supports available to their child who consistently scores in the upper percentile. Because they are the ones actually delivering the curriculum, HEI faculty have considerable sway over prospective students [112,140], as they are in a prime position to disseminate information about the HEI's overall characteristics [138]. When they get to know one other better, students begin to feel more confidence, comfortable and safe in their academic decisions [141].

## Preferences for information sources

Prospective students can get a head start on the evaluation process by perusing the HEI's controlled printed resources, such as prospectuses, brochures, and leaflets [142–144]. Students can utilize these resources as a jumping off point [99] and to learn more about the academic programmes, geographical amenities, and cost of living associated with HEIs [38]. A number of studies have voiced concerns that the publications lack in credibility [11], provide misleading information [145], or are otherwise unethical [146]. It is for this reason that prospective students do not only depend on these resources on arriving concluding decisions about deciding which EC to enroll in. Whether on or off campus, prospective students may get their doubts cleared by HEI staff. Prospective students are fascinated by face-to-face communication by means of counselling seminars or campus tours [147,148]. Inspiring, productive, and providing confidence, resourcefulness like this fosters a vested interest in deciding which HEI to attend [142,147,149,150] Prospective students consider HEIs' websites to be a persuasive tool [147,151–154]. They found it easy to read and understand with detailed content [155] that exhibits a wide range of specific information on current events, campus life, and placements; details they would not have been able to obtain in a timely manner from other sources within HEIs. Prospective students are comfortable utilizing social media to attain their educational goals [156] because of its familiarity and the quick information it delivers [157,158] with no time and cost constraints [159]. Social media is a powerful tool for fostering two-way interactive communication [124] among prospective students, existing students, alumni, and other stakeholders that houses continuous engagement and relationships in order to foster long-

term reliance [160]. Social media is a valuable tool for collaborative and creative work that encourages long-term collaborations [161] while effective branding tool in the co-creation of "word of mouth" leading students towards HEIs [162].

## College's characteristics

College choice sets reflect students' aspirations required for the evaluation of selecting a HEI itself. These characteristics, which stand in for the HEIs facilities and services, are divided into three categories: financial versus non-financial, academic against non-academic, and tangible versus intangible [163,164]. The study of [165–167] refers them as service mix and brand enhancing strategic tools [168] of HEIs that has a potential to fulfill HEIs' objectives of enhancing students' enrollments, performance and satisfaction.

## Proximity

The distance between hometown and HEIs is referred to as proximity. Small-distance travel saves money for family, time allows more interactions with friends and effort lets extended homework / study hours [11], which are the crucial advantages of proximity. Consequently, nearness to HEIs is a crucial influencer in college choice [169,170], especially for females [171] and lower SES groups [172]. It also boosts the likelihood of chance of students' college going [173].

## Location and locality

Locality refers to the culture, amenities, and facilities in the region surrounded to HEIs, whereas location relates to its position and accessibility from home. Location and locality is not limited to physical borders but can mean different things to different stakeholders and be visualized by the beholder [174]. The structure of ambient conditions, physical layout, and functionality [175] of location and locality affect college choice decisions [143,176–179]. Its appropriateness, convenience, attractiveness, accessibility, cost-effectiveness safety, and security are counted [180,181]. Safety premises for females and affordability for low-income students are the concerns about location and locality [182]. EE relies on physical and digital location like online learning platforms and social media [183] to engage stakeholders. In India, a country with numerous languages and cultures where the humanities alter every shift, students are more hectic in considering geography and neighborhood [165].

## Image and reputation

Public perceptions about image and reputation is a crucial differentiator [184] though it is intangible but has been identified as one of the most influential elements in HEI choice decisions [185–187]. Academic staff reputation, brand status determined by league tables or newspaper, national ranking, gradation, quality and HEIs lively presence are all factors that counts for students impression [168,188]. To be sure, it is a complex spectrum of various reputes such as quality facilities and services academic or otherwise. Even if no one ever encounters a HEI in person, the reputation and public profile will make an indelible imprint on the minds of individuals making word of mouth and choice decisions for HEIs [148,185,189,190].

## Faculty

HEI's faculty have a significant impact on students in terms of their size, qualifications, competency, tenure-line and experience [184,185,191–194]. A robust foundation of faculty attributes and attitude is required for successful engineering curriculum delivery [195] as their

interaction with students produces substantial positive impact [183]. Faculty with high-quality teaching [196] and attitudes [197] are almost guaranteed to be recognized. They, too, must be highly motivated, knowledgeable, enthusiastic, adaptable, and responsive [198] in order to effectively communicate and guide students through meaningful experience learning [199].

### Alumni image

Alumni of HEIs are their finished products, having benefited greatly from their educational services. For of this, alumni's accomplishments and social status are often used to demonstrate the significance, eminence, quality, and brand image of HEIs and to provide preferential grounds for choosing a HEI [123,200,201]. Because alumni are the tangible outputs of educational services, students and their parents take support from alumni to evaluate HEIs performances. Alumni networks are continuously used to encourage new enrollments because of the positive impact they create on society. Renowned alumni are unquestionably valuable resources for HEI brand recognition [202]. Alumni with market and social stature set benchmarking norms for prospective students [203].

### Campus placements

Job opportunities or employment prospects are potential outcomes and benefits of HEIs [111,124] that prospective students and their families desire to know about [189]. The potential of a rewarding employment through a placement cell is a major consideration for the students [191,204], especially for male [205]. Most ECs have placement offices whose mission it is to improve relations between industries and HEIs in order to better prepare their students for the employment [206]. Campus placements can be promoted through expanding employment prospects to improve employability skills [207] and accelerating industry-academia connections [204].

### Quality education

To remain copetitive and gain credibility among its stakeholder group, ECs focus on the quality academic offerings. The quality factor, which is responsible for the qualitative and holistic development of students is crucial in deciding HEI selection [14,176,208–210]. Over the years, service quality in HE has been the topic of debate for several research, including [23,211–213]. In India, the quality of HEIs is judged by a number of factors; academic infrastructure and services, connections to the corporate world, on-campus jobs, research, accreditation, and financial stability [178]. Quality issues in HE are acknowledged in the previous research in regards to academics and extracurricular [213,214] curriculum [215], delivery of courses [216], infrastructural facilities [217], faculty and staff [218], and the provision of support services [219], Ultimately, quality has both concrete and intangible consequences for the selection of ECs [220]. High-scoring students [221], eminent faculty [20], a reputation [222], and positive recommendations from the existing enrolled and alumni students [223] are all indicators of a high-quality of ECs.

### Infrastructure and facilities

The importance of infrastructure and facilities has been emphasized by a number of studies, including [224–226] as one of the factors considered in college choice [227]. It is a tangible asset includes the physical infrastructure such as buildings, furniture, equipment, and computational and IT infrastructure [228]. As first impressions are frequently the most enduring, all stakeholders respond to its tangibility in emotional and physiological ways, which in turn

influences HEI choice actions. In order to make a quick choice, a good tangible resources must be provided by HEIs [147,152] Curriculum delivery in EE is not feasible without it because of the technological nature of EE. More students are attracted towards if the infrastructural provision is greater, but the cost associated with it is a key consideration.

## Safety and security

Campus safety refers to the steps taken to assure students' well-being in regard to their residence, health, and everyday life [57], whereas security in its fullest meaning involves physical, social, and economic elements through defending human rights, emotions, and cultural values [229]. Numerous studies like [191,230,231] have shown that students consider it from the perspectives of their own well-being sought for humanizing culture, ethics and values, while making a college choice decision [57]. Students have high praise for the college's health services, sanitation services, fire safety, and emergency measures [232]. Improved physical and mental health may be achieved more comprehensively via the implementation of safety protocols and regulations [233].

## Curriculum delivery

Curriculum delivery is the backbone of academic in EE [99,131]. EE has traditionally been content-centered, hands-on, design-oriented, and focused on developing critical thinking and problem-solving skills [35], but it encountered a controversy due to its inability to encourage professionalism [234], engross a practical and application-approach [235], industry exposure [236] and to attract students from diverse backgrounds. Although each approach has advantages and disadvantages in theoretical, practical, technological, human-involvement, and skill-based approaches, what matters most is how well it is processed knowledge among diversity to acquire skills [237], make it substantial [109], interdisciplinary [108], and inclusive [238].

Online learning has opened several possibilities since it can be done at any time and from any location [239]. Online learning is an appealing choice because to its flexibility and ease [106], but the usage of computers [240], cutting-edge online technology, such as ICT [241], and instructor motivation and attitude [242] are recognized matters for its success. It is not ideal and sustainable for EE due to logistical, technological, and learning/teaching concerns [243] as well as its incapacity to illustrate specific technical or mathematical subjects [244]. A number of studies have proposed hybrid/blended delivery [105,245] as a viable answer to continuing education, but these debates have lacked an analytical base. According to [246], this causes a loss of identity, control, and interrupted involvement. In engineering, onsite delivery [104] is critical to maintaining academic focus and motivation, which provides actual justice for becoming a capable engineer. Curriculum delivery was considered to be the top consideration when selecting ECs [46], as it is recognized to transform engineering knowledge into practical and technology applications [247] and attract diversity [248]. With this knowledge, students as well as ECs are ever more concerned about the nature of curriculum delivery that is more sustainable during COVID-19 pandemic and preserves the value of engineering.

## Value for money

The perceived financial burden about HEI costs influences students' enrollment. Cost involved in terms of tuition, conveyance, educational equipment and accommodation, related with ECs are disproportionately expensive in comparison to other HEIs. If fewer families can afford to invest in their children's EE, the number of students choosing ECs may fall [249]. However, literature suggest, the influence of college costs in enrollment decisions has been reduced over time [250]. India, cost of education is the first prima facie criteria for lower SES, and female to

make the decision of selection of ECs [251]. It has an impact not just on the ECs' income, but also on student judgments of value for money and service quality [252]. As the value of an education at a particular HEI grows, the cost curve may shift in the other direction, making that HEI the better option. [194]. Several studies are of the opinion that cost of education incurs rewarding advantages, such as value and service quality [176,253,254], time and effort [255], effort and opportunity [256], high earning benefit [154], and value for benefit sought [181,257–260].

## The changeover in the context to pandemic: Base for conceptual framework

**A flash back to helter-skelter-like situation in India.** COVID-19, a coronavirus-caused respiratory disease conveyed by intimate human contacts, was discovered first in December 2019 [261,262]. Due to the disease's widespread nature, the first priority for HEIs was to take precautions for their in house students' safety as learnt by Maslow before Bloom philosophy. The disease was spreading like wildfire over the world, creating the most dreadful and unimaginable working conditions for HEIs. The only suitable means of avoiding its progression, according to the World Health Organization (WHO) was to induce physical and social distancing that gave impression to educational long-COVID state.

During pre-pandemic period, India's traditional HE system was based on face-to-face interactions between teachers and students to impart knowledge and skills, a concept known as offline learning. The first rumblings of HEIs closures in India occurred on March 25, 2020. Officials of Govt. of India, to restrict the infection spreading, suspended 'onsite' physical workouts and shifted them 'online' [263], the effectiveness and performance of which was confirmed during previous natural disasters [264]. To sustain physical health, HEIs were subjected to abrupt shifts by switching to digitalized platforms overnights. This sparked a digital revolution in the education industry, necessitating significant changes. All Indian HEIs were rigorous on online teaching and learning to complete the remainder academic of 2019–2020 for existing students who were already enrolled in 2019–2020. This abrupt shift to online education, initially created various arguments including 'digital challengers' [263,265], 'continuing education further' [242,264,266–268] and 'remodel in the midst of a crisis' [269]. As the online approach became widespread in the middle of 2020, it was critiqued for a number of reasons, including students' ability to focus, interest, motivation, quality, access, interactions, digital connectivity, and support system [28,270–279] and many more. These were concerns for ECs, as EE was generally delivered through a integration of lectures, tutorials, laboratory practical, team work, and fieldwork [102,280]. Moreover, studies reported students' anxiety, depression, and other mental health problems as a result of the COVID-19-related isolation from peers and friends, campus socialization and support systems [281–283].

Overall, COVID-19's extraordinary conditions prompted rapid changes in EE while also questioning the status quo's assumptions and structure. Nevertheless, in India, homogeneous conducts of leader towards IT approach [284], elastic approach and the initiatives of UGC and AICTE in regard to online EdTech Start-ups [285] and availability of private but paid digital platforms [2] had shown substantial potential for a diverse student population enrolling in HEIs [72]. There still, most of HEIs were unprepared for the rapid pace with which the shift to online learning was anticipated to be carried out. ECs, prepared or not, were pushed to complete the remaining pedagogy by the end of May 2020. Not all Indian ECs had been successful in transitioning from offline to online education. As a result, statistics revealed that students were failed to perform normally and to acquire knowledge at the same rate as before. This disparity was highest for disadvantaged groups [72], raising concerns about the digital transformation's long-term sustainability. Additionally, the pedagogical elements linked with face-to-

face mode in pre-pandemic were replicated same in online mode as they were present before. This kind of carryover was not was not expected considering mental wellness of the students [286,287]. The need for reframing the policies was immediately recognised in order to protect students' psychological well-being [288], which was required essentially at that time for students' better academic performances [289]. In this view, to retrieve psychological sentiment, an MCQ and OMR-style online exam was administered and conducted in Jun/Jul 2020. The results of which were made public in Aug/Sep 2020, based on the evaluation of students' performance in the online examination and their earlier internal or semester assessment available. In terms of students' dropout in 2020, it was found that females, rural students, and those from lower SES origins exercised major impacts due to academic, social, and emotional losses. Various new procedures have been proposed to ensure that engineering students do not slide down the track [290]. Others have proposed setting-up small campuses in rural and remote regions, where adherence to social distancing may be easier to administer education activities physically, than the decimated metropolitan areas [291], however beheld as not feasible.

As the intensity of COVID-19 lessened, the unlocking phases encouraged Indian ECs to reopen their campuses in physical mode in November 2020 with the priority of continuing education in academic year 2020–2021 for the existing students who were promoted from the previous academic year 2019–2020, in accordance with guidelines issued by the MHRD, India, and the UGC. Upon the reopening of the campuses, regulating authorities and ECs together collaborated successfully that assured existing students had adequate opportunities to continue their studies further. The most difficult part of recommencing was to ensure pandemic safety measures for offline learning with safeguarding 50% enrolments every day, phase by phase and batch by batch. Before December 2020, the EC's cardinal topic was to increase knowledge levels and ensure the steady academic and to stop drop-out of existing enrolments. As ECs and existing students began to settle down, the offline pedagogy pressure started to ease. Owing of pandemic constraints, the ECs administration's approach to emergencies turned substantially to upkeep the morale of existing students.

As they groaned at the prospect of accomplishment by the end of Dec 2020, ECs administrators confronted a new challenge: attracting and admitting new enrolments (new aspiring entrants) for the academic years 2020–21, which had not previously been on the priority list of ECs. ECs administrators, who started predicting fresh enrolment (2020–2021), their marketing strategies had been blocked for significant statistic indicating drops in enrolments due to widespread apprehension about ability of families to afford ECs in the wake of job losses, meeting the cost of education, soaring unemployment, and the prospect of enrolling in online stirred program [292]. Even though they developed a marketing campaign for prospective enrolments, they appeared to have missed out the window of opportunity counting on time frame and potential targets. Despite their best efforts, they were unable to create attention, interest and demand for their ECs' promotional offers, as the concentration of potential target was already skewed towards the "larger picture" of pandemic concerns [293].

While this all has happening in India, statistics of Europe revealed a decline of 6 to 10% over forth coming years which surely was not unfavorable for Indian HEIs. According to research reports, the effects of the pandemic speckled greatly by SES position [294]. Along with affecting emotional and personal life, the pandemic has had an impact on demography, socioeconomic features, academic achievement, and economic stability [295], resulting in a shift in priorities for ECs as well as prioritised offers of ECs [296]. Yet, some evidence indicated that families were well prepared to spend in HE as they had intended before to the pandemic [297], making a pleasant news for ECs. Overall, analysts were concerned about the continued losses caused by a shift in attitudes about decision of EC choice, which looked to threaten an economic trajectory.

Though prospective students of Gen alpha were not directly involved tangibly, they including their influencers were the witnesses of the recent volatile scenario and happenings linked to ECs. But then again they were the experience holders of the educational impact created by this pandemic when they were attempting their higher secondary school at the start of pandemic year. As a result, we attempted to explain broadening this chaotic and helter-skelter-like condition, which we believed might have created an impact on aspiring students' intentions towards the sustainability and current hazards linked with ECs. Gen alpha as per its signposts are hypothetical to be more sceptical of the characteristics, attitudes, and expectations involved in 'college choice' decisions.

The admission process for new enrollments for the first year of a four-year undergraduate engineering degree programme for the academic year 2020–2021 was carried out during January 2021 under the supervision of the state competent authority, which was nearly six months late to earlier pre-pandemic years. With the presence of the COVID-19 pandemic, ECs welcomed newly enrolled students on-site in February 2021 to steer them towards their dream career they desired, following the pandemic guidelines.

## Emergence of 'sustainability' during pandemic

There is little consensus on a single meaning of the term "sustainability", which has evolved over time and across different fields of study [298,299]. Very few studies like [108,300] addressed the need to embrace sustainability in EE before the pandemic situation with different approaches. Sustainability in education is all about designing practices that can be scaled or right-sized without depleting resources or exclusion of groups [301]. Sustainability became more important in higher education right before the outbreak of the pandemic, with HEIs developing customised medium- and long-term strategy plans focusing on sustainability. However, the COVID-19 pandemic has halted several initiatives and strategic plans. Most HEIs were compelled to adjust their focus from 'sustainability' to 'survival' as a result of the pandemic influence, blocking the path of transformative change. If sustainability is to be revived as a development objective, then during pandemics, the prerequisite condition for HEIs is the development of effective plans and policies that attract new enrolments, fill seats to capacity (full resources), and raise revenues, without which sustainable system may be hindered to obtain desire effects [302]. As sustainability relates to the capability of being supported or maintained or kept going [303], it must be reviewed during pandemic.

The approach of this study towards sustainability is to employ it to withstand changing attitudes and performances [304] such as students' attitude affecting new enrollments, ECs 'revenues and performances that needed to cope with the unexpected environmental challenges [305] like COVID-19 pandemic impact on EE over the future [306] i.e. as long as the pandemic exists. Aspiring students and their influencers are now more suspicious and hesitant and to make "college choice" decisions under the global pandemic, as evidenced by the literature and statistics of the reports on decline enrollments. As a result, we tried to establish a new term, "sustainability," to describe the ideal convergence between aspiring students' expectations and ECs' provisions in the event of a pandemic.

After analyzing the literature and noting the effects of the COVID-19 pandemic period (Dec 2020–Feb 2021) on ECs, we circumscribed 'sustainability' from the point of view of including new fresh enrollments. Here, sustainability is to maintain the pre-pandemic performances ongoing as long as pandemic be existent, in terms of availability of resources and services related to wellbeing, the delivery of EE and the development that the aspiring diverse students require, without being affected by the impact of pandemic.

Thus in short term, addressing sustainability of ECs refers to the provisions made under pandemic supporting;

- pandemic measures [307,308]–that ensures safety, security and to protect health

- ethical engineering [309]–that establishes technical, professional, and social sphere

- inclusive curriculum [108,238,310]–that incorporates appropriate curriculum making suitable for pandemic situation protecting competency of engineers

- diversity, inclusion and equity [70,73,310] that enables curriculum delivery for all

- psychological well-being [283,289]–that maintains mental strength

- **Research gap and significance of study**

No studies have undertaken initiatives to reflect the effects of sociological and economic aspects on college choice decisions during a pandemic situation. The vast majority of research works published at the time of the pandemic seems to be concerned with maintaining and establishing academic governance for the students already enrolled. Throughout the pandemic, "college choice" was not attempted, as far as we are aware. Future research was required to delve into the complexity of choice characteristics in order to better understand the compatibility of choice characteristics associated with both; prospective students and ECs in the situation of a COVID-19 pandemic.

Second, many of us feel that the attributes of diversity have altered substantially as a result of a compelling pandemic. For this reason, it is vital that ECs reorient the policies that will support the situational change necessary to sustain future enrollments by understanding the progression of choice characteristics that investigates the impact of the pandemic on future enrollments. As the future of ECs' budgets and the nation's economy hinges on the new enrollments, this issue needs to be addressed immediately.

The intent of our study is to bridge this knowledge gap by analyzing the choice characteristics that sustain students' decisions about which EC to enroll during a pandemic, making it both contemporary and relevant. Secondly, it uses the framework to evaluate the relationships among choice characteristics facilitating the diverse enrollments into ECs.

## Conceptual framework and hypothetical model

College choice process transpires concurrently for both sides; aspiring students as well as ECs. One side of students are concerning about which EC to enroll that gives perfect match to their personal traits. The other side of ECs relates to the offers influencing students' choice [311].

This study corresponds to the "choice" stage, where students are encouraged to enroll in an EC from the available priority list. After gathering sufficient information and being motivated by human network, students usually evaluate and rank ECs to attend that is best fit for their needs and benefits [181]. Prospective students, in consultation with their friends, family, and teachers, obtain opinions and suggestions. The next step is to select an EC that is well fit to sustain learning EE during the pandemic. Finally, the collective decision of 'college choice' is based on the appealing and useful offerings made by the ECs, which include tangible and intangible attributes. These attributed are collected by the prospective students and their influencers from ECs marketing tools and informational resources, in an effort to sway their decisions. Such a data/information is obtained from sources may either online or offline, interactive or non-interactive they believe to be most relevant and trustworthy. Yet, the COVID-19 pandemic calls for a more intellectual and rigorous screening of ECs characteristic —sustainability, that most appropriate to evaluate to meet the needs under pandemic. The

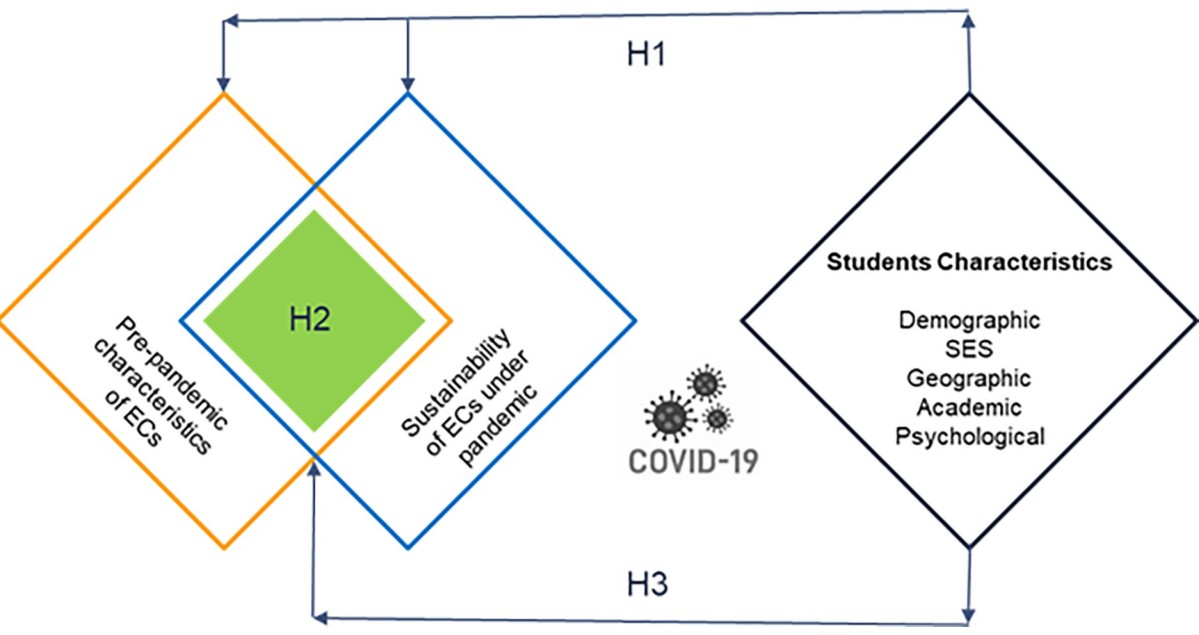

**Fig 1. Choice characteristics under COVID-19 pandemic.**

following hypothetical model (Fig 1) is built to meet the objective and to answer research questions of this study based on the theoretical and conceptual framework as stated previously. Also, the following three hypotheses associated with research questions have to be validated based on students' perceptions.

Hypothesis one (H1)

There is significant difference in students' perceptions of sustainability of ECs under the COVID-19 pandemic.

Hypothesis two (H2)

There is significant relationship between the sustainability of ECs under COVID-19 pandemic and the ECs' characteristics.

Hypothesis three (H3)

There is significant relationship between the sustainability of ECs under the COVID-19 pandemic and the ECs' characteristics within and across students' characteristics.

## Research methodology

### Setting the research domain scene

Being educational research, this study focuses on a predicament faced by engineering students deciding EC choice, especially acute in light of the COVID-19 pandemic. A literature review aligned with the objective of this study has enabled this study to implement a quantitative method due to its ability to frame hypotheses [312], capabilities to operate on multivariate statistical data [313], ability to analyze relationships with definiteness and transparency [314], reliability [315] and being successful in educational research [316]. Since the 'student' is the primary 'customer', their opinions should be monitored frequently [317]. Students' perceptions of their service encounters are often used as the basis for making value judgments about whether or not those encounters fulfilled students' expectations [318]. This research relies heavily on the students' own perceived experiences. Students' perspectives about their pathway they took to select an EC after the pandemic will be reported through this research.

Since this study of college choice is solely related to the influence of the COVID-19 situation which has an existence in India from March 2020, students who have had experience of 'college choice' before March 2020 i.e. who were enrolled in or completed engineering studies before pandemic cannot be viable demographic for this study and hence are not considered. The research population involved exclusively of fresher engineering students who have an encounter of 'college choice' after the influx of pandemic in India were selected purposefully and hence making perfect population fit for the study. Purposive sampling has been chosen decisively because of the knowledge and judgment of researchers [313,319], special situations [319] and investigations of new issues [320] about 'college choice' during the COVID-19 pandemic.

The sampling frame for this study consists of thirty nine engineering colleges situated in North Maharashtra, India that offer bachelor's degrees in engineering and technology. Sampling units consisting of fresher students representing batch of academic year 2020–201 who had selected their ECs in the pandemic month of Jan 2021, were chosen from a sampling frame to cover a variety of demographic, geographical and socioeconomic characteristics of the population. After sending a request email, the ECs involved under study provided 4,300 contacts involving e-mail addresses and WhatsApp numbers of required students. An online self-report survey [321] was conducted over the internet with Google Form tool for better flexibility in time and place, and to make students more responsive and quicker. A Google Form tool link containing the survey questionnaire was circulated through E-mails and WhatsApp to the contacts received above in the month of Feb 2021. Self-reported survey was supportive during the COVID-19 pandemic because it eliminated the possibility of physical contacts and researcher bias, but ensured maximum reach in the stipulated time. The questionnaire link was discontinued before the start of academic sessions for new enrollments in the middle of Feb 2021. The survey obtained 516 valid responses on the researchers Google drive who only had the access to the research data, with a response rate of 12%. A sample size of 516 for assessing eleven variables, with a ratio of 47 samples per variable, is sufficiently defensible for Regression analysis in statistics, which entails a minimum of 10 to15 samples [322–324] [325] per variable. Out of 516 valid sample population who were enrolled in first year, there were 348 male (67.4%) and 168 female (32.6%). Among these 123 (23.8%) and 393 (76.2%) belonged to lower SES and higher SES respectively. In case of geography, 193 (37.5%) were from urban areas and 323 (63.6%) were having rural native place. Out of 516 accepted samples, 386 students were enrolled to computer allied and 130 belonged to non-computer allied majors. The age rage of all was in between 17 to 19 years.

## Data collection

Primary data were gathered by a quantitative survey method that deemed optimal for eliciting opinions from a large sample of respondents [326]. A quantitative survey was administered with a list of structured closed-ended questionnaires based on literature survey and online informal discussions held with the students who were enrolled in the ECs during pandemic and processed according to the guidelines of [327] and [328].

Section I of the questionnaire provided context for the study and explained why collecting this data was important. At the bottom of this section, everyone's approval on participating in the survey and using their responses in survey results in a public setting was requested. As this survey was voluntarily and anonymous, the participants had a choice not to participate into it by simply declining the request and withdrawing themselves from the survey without any consequences. After the participants gave their approvals, the following segment (Section II) was permitted to proceed. The Section II asked students about their five self-related characteristics

which were the independent categorical variables Q1) demographic characteristics: gender, Q2) socioeconomic characteristics (SES), Q3) geographic characteristics: native place, Q4) academic characteristic related to higher secondary academic performance (pre-college performance) Q5) the psychological and behavioral characteristics. It further constituted four sub-questions related to students' information about Q5-1) engineering major enrolled in Q5-2) priority for type of curriculum delivery Q5-3) the most valuable human influence, Q5-4) the most effective informative source. To answer questions Q1 to Q5 included in Section II students were asked to choose one choice from a drop down list (sub-items) of options linked with the question. Such sub-items can be understood from Table 2 under the column- sub-items for grouping.

Sections III encompassed Questions Q6 to Q17 related to twelve ECs characteristics in including proximity, location and locality, image and reputation, faculty profile, alumni profile, campus placements, quality education, infrastructure and facilities, safety and security, curriculum delivery, value for money and lastly sustainability under COVID-19 was examined. Section II encompassed eleven continuous but independent and one continuous but dependent variable (sustainability). To answer questions Q6 to Q17 included in Section III students were asked to rate on a Likert scale (1 to 5), where 1 represented minimum value and 5 showed maximum value. For better understandability of the questions included in section II and III, descriptive narration was provided to the questions.

Sections II and III were intended to address research questions; RQ1 and RQ3 worked together, while Section III was created to solve research question RQ2. All of these sections including seventeen questions were organized in a logical sequence. To validate hypothesis H1, Analysis of Variance (ANOVA) was utilized to identify the significant difference in students' perceptions of sustainability of ECs under the COVID-19 pandemic. Hypothesis H2 was tested by Regression Analysis to detect the accountable relationship in between the sustainability of ECs under COVID-19 pandemic and the ECs' characteristics. ANOVA was performed to predict the statistical significant across the students' characteristics to explain regression model representing the relationship in between the sustainability of ECs under COVID-19 pandemic and the ECs' characteristics to justify Hypothesis H3. All above statistical analysis were performed by Statistical Product and Service Solutions (SPSS).

## Ethical considerations

Before entering the actual survey, the validity and reliability of the questionnaire was tested through pilot testing [329,330]. Forty valid samples corresponding to sampling units were selected to understand its suitability for conducting the online survey. Their feedbacks were considered for correcting the technical language and sequence of the questionnaire.

Online permission was obtained from all ECs included in the sampling frame in order to use data they provided in the administration of the self-report survey. Next, the Institutional Ethical Committee of R. C. Patel Institute of Pharmaceutical Education and Research, Shirpur (India) had given permission to conduct online survey and to obtain online consents of human participants as per the letter RCPIPER/IHEC/2021/004, dated Jan 14, 2021 that confirmed the study followed all applicable national guidelines and protocols for research involving human subjects.

## Statistical results

To explore the relationships between the sustainability of ECs under the COVID-19 situation (CS, dependent and continuous variable), college characteristics (C1 to C11, independent and continuous variables) and student characteristics (S1 to S8, independent and categorical),

SPSS 25 is applied along with the guidelines of [331]. For simplicity students' characteristics (S1 TO S8) are formed by grouping of similar featured sub-items representing concurred characteristic as presented in Table 2.

## Statistical tests and fitness of data

To find the solution to the research question RQ1 and validate H1, ANOVA was conducted as directed by [332], which demonstrated the association of independent variables S1 to S8 (categorical, student characteristics) with the dependent variable CS (continuous, college characteristics). Regression analysis was considered as it is the most preferred and useful technique in assessing educational behavior, where the responses are quantitative and continuous [333,334]. Simple regression was performed exploiting continuous variables C1 to C11 as a predictors CS associated with college choice decisions to answer RQ2 and justify H2 [335]. After that, to resolve RQ3 and H3, the continuous variable CS was analyzed to predict the portion of its variance on account of the variations in continuous variables C1 to C11 across the categorical variables S1 to S8 variables one by one.

A multicollinearity test in SPSS was conducted to confirm the existence of significant correlation among a group of independent variables. The test demonstrated that tolerance >0.2 and VIF values <10.0 are well within the specified limits. It ensured that the data had no issues with collinearity and standard errors associated with the predictors [336–338]. Next, reliability based on internal consistency validated uniformity for all independent predictors comprising variables C1 to C11 and CS (continuous, college characteristics) with coefficient alpha [339,340] value of 0.941 for the entire group (>0.6). These values expressed the best fit for study purpose [341] (Refer Table 1). Based on the corrected item-to-total correlation obtained, as shown in Table 2, positive values of α ranging above 0.3 indicated good internal consistency of the scale [342]. Last, Tukey's test detected no additivity that confirmed a sufficient estimate of power.

**Table 1. Descriptive statistics and reliability of college characteristics.**

| Question | College characteristics | Code | Mean (μ) | Standard Deviation (σ) | N | Corrected Item-Total R | $R^2$ | Cronbach's α if Item Deleted | Rank in contributing to EC choice |
|---|---|---|---|---|---|---|---|---|---|
| | Cronbach's α for Group | | | | | | | 0.941 | |
| Q6 | Proximity | C1 | 3.480 | 1.321 | 516 | 0.346 | 0.297 | 0.937 | 12 |
| Q7 | Location and locality | C2 | 4.120 | 0.826 | 516 | 0.719 | 0.621 | 0.920 | 9 |
| Q8 | Image and reputation | C3 | 4.140 | 0.784 | 516 | 0.778 | 0.718 | 0.918 | 3 |
| Q9 | Faculty | C4 | 4.060 | 0.794 | 516 | 0.805 | 0.715 | 0.917 | 4 |
| Q10 | Alumni | C5 | 4.020 | 0.851 | 516 | 0.766 | 0.654 | 0.918 | 7 |
| Q11 | Campus placements | C6 | 4.070 | 0.812 | 516 | 0.753 | 0.663 | 0.919 | 6 |
| Q12 | Quality education | C7 | 4.080 | 0.774 | 516 | 0.840 | 0.756 | 0.916 | 2 |
| Q13 | Infrastructure and facilities | C8 | 4.020 | 0.785 | 516 | 0.765 | 0.628 | 0.919 | 8 |
| Q14 | Safety and Security | C9 | 4.080 | 0.760 | 516 | 0.798 | 0.686 | 0.918 | 5 |
| Q15 | Curriculum delivery | C10 | 4.020 | 0.797 | 516 | 0.836 | 0.761 | 0.916 | 1 |
| Q16 | Value for money | C11 | 3.870 | 0.928 | 516 | 0.707 | 0.539 | 0.920 | 10 |
| | Covid-19 sustainability | CS | 3.640 | 1.049 | 516 | 0.632 | 0.435 | 0.924 | 11 |

Source: Reliability test run through SPSS

## Characteristics of ECs leading to college choice

Table 1 presents descriptive statistics of students' perceptions about ECs characteristics wherein they enrolled, along with squared multiple correlation ($R^2$) and Cronbach's α. The overall Cronbach's α for all college characteristics (C1 to C11 and CS) is 0.941, indicating strong strength about scale uniformity and reliability towards measuring college choice. Image and reputation (C3, μ = 4.140, $R^2$ = 0.718), followed by location and locality (C2, μ = 4.120, $R^2$ = 0.621), quality education (C7, μ = 4.080, $R^2$ = 0.756), Safety and security (C9, μ = 4.080, $R^2$ = 0.686) and campus placement (C6, μ = 4.070, $R^2$ = 0.663) are among the top EC characteristics that have enchanted college choice decisions for the students. However, proximity (C1, μ = 3.480, $R^2$ = 0.297) and sustainability under COVID-19 (CS, μ = 3.640, $R^2$ = 0.435) were found at the bottom of the list that constituted important characteristics controlling college choice decisions.

## Sustainability across students' characteristics (H1)

Descriptive statistics of eight independent categorical variables (S1 to S8) representing students' characteristics and results of ANOVA test exhibiting its association with sustainability under Covid-19 (CS) is presented in Table 2.

The results of ANOVA displaying univariate analysis showed that gender (S1) did not have any impact on sustainability under COVID-19 (CS, $F$-value = 2.879, $\rho$ = 0.09) was statistically insignificant at $\rho > 0.05$. This destined that there were similar perceptions between male and female groups regarding the sustainability of an EC. Similarly, SES (S2, $F$-value = 0.998, $\rho$ = 0.318), native place (S3, $F$-value = 1.052, $\rho$ = 0.306), and school performance (S4, $F$-value = 0.453, $\rho$ = 0.501) did not show any statistical significance ($\rho > 0.05$), indicating that there were similar views on CS across S1, S2, S3 and S4. Hence, the H1 was rejected and alternative hypothesis ($H_01$) was accepted for S5, S6, S7 and S8, showing no significant differences with CS.

When college sustainability under COVID-19 (CS) was assessed with students' major choice (S5), significant disparities with $\rho < 0.05$ were observed across the group (S5, $F$-value = 8.333, $\rho$ = 0.004). The students who opted computer-related major courses were observed having greater attention to CS (μ = 3.72) than the students who had chosen non-computer majors. With $\rho < 0.05$, course delivery mode (S6, $F$-value = 25.601, $\rho$ = 0.000) had the greatest effect on sustainability under the COVID-19 situation. CS appeared to be the greatest concern for those preferring onsite learning (μ = 3.92) and hybrid learning (μ = 3.48) than the students emphasizing online learning (μ = 3.19). There was a significant difference ($\rho < 0.05$) between CS and human influence (S7, $F$-value = 2.686, $\rho$ = 0.046) responsible for students' college choice, Family had greater influence (μ = 3.84), followed by college representatives (μ = 3.69), community (μ = 3.52) and pre-college (μ = 3.43). Information sources (S8, $F$-value = 2.560, $\rho$ = 0.049) had statistical significance ($\rho < 0.05$) with CS. A Non-interactive-online information sources (μ = 3.77), was proven to have the greatest impact in providing information about CS, followed by Interactive-offline (μ = 3.71), Non-interactive-offline (μ = 3.67), Interactive-online (μ = 3.46). Hence, the hypothesis (H1) was accepted for S5, S6, S7 and S8, where there was a significant difference noted with CS.

## Relationship of sustainability with ECs characteristics (H2)

The regression model (refer Table 3) showed that the relationship between eleven independent characteristics of colleges (C1 to C11) and the dependent variable (CS) is statistically significant ($F$-value = 35.220, p<0.000) in predicting the relationship CS←(C1 to C11). It further confirmed that the model is associated with a large coefficient of correlation ($R$ = 0.659) [343]

**Table 2. Sustainability under Covid-19 based on Students' characteristics.**

| Question | Students Characteristics (S) | Sub-items for grouping | Sustainability under Covid-19 (CS) | | | | |
|---|---|---|---|---|---|---|---|
| | | | N = 516 | μ of CS | F-value | ρ | Hypothesis Support |
| Q1 | Gender (S1) | | | | 2.879 | 0.09 | $H_0 1$ |
| | Male | Male | 348 | 3.59 | | | |
| | Female | Female | 168 | 3.76 | | | |
| Q2 | Socio-economic status SES (S2) | | | | 0.998 | 0.318 | $H_0 1$ |
| | Higher social class | General class Other backward class | 393 | 3.76 | | | |
| | Lower social class | Backward class | 123 | 3.56 | | | |
| Q3 | Native place (S3) | | | | 1.052 | 0.306 | $H_0 1$ |
| | Urban | District, Taluka | 193 | 3.70 | | | |
| | Rural | Village | 323 | 3.61 | | | |
| Q4 | Pre-college performance (S4) | | | | 0.453 | 0.501 | $H_0 1$ |
| | High scorer | XII standard marks >75% | 210 | 3.68 | | | |
| | Low scorer | XII standard marks <75% | 306 | 3.84 | | | |
| Q5-1 | Major choice (S5) | | | | **8.333** | **0.004** | **H1** |
| | Computer allied courses | Computer, Artificial Intelligence, Machine learning and Data Science | 386 | **3.72** | | | |
| | Non-computer allied courses | Mechanical, Electrical, Civil and Electronics | 130 | 3.42 | | | |
| Q5-2 | Course delivery mode (S6) | | | | **25.601** | **0.000** | **H1** |
| | Online | Online | 132 | 3.73 | | | |
| | Hybrid | Hybrid | 106 | **3.92** | | | |
| | Onsite | Onsite | 278 | 3.48 | | | |
| Q5-3 | Human influence (S7) | | | | **2.686** | **0.046** | **H1** |
| | College | College staff and college students, alumni | 225 | **3.69** | | | |
| | Community | Friends and relatives | 150 | 3.52 | | | |
| | Family | Family members | 90 | **3.84** | | | |
| | School | School/Jr. College | 51 | 3.43 | | | |
| Q5-4 | Information source (S8) | | | | **2.560** | **0.049** | **H1** |
| | Non-interactive-offline | Media advt., banners, hoardings, leaflets | 81 | 3.67 | | | |
| | Non-interactive-online | Website | 162 | **3.77** | | | |
| | Interactive-offline | Seminars, exhibitions, campus visits | 114 | **3.71** | | | |
| | Interactive-online | Social media | 159 | 3.46 | | | |

Source: ANOVA run through SPSS.

Note: Values in bold text are significant at ρ<0.05.

and coefficient of regression ($R^2$ = 0.435) having moderate to strong strength [333,344,345] in predicting CS, even though the data narrated unpredictable human behavior. This model further guaranteed that CS significantly predicted 43.5% of its variance, accounted for by independent predictors (C1 to C11). It is also observed that ECs characteristics, i.e. sustainability under Covid-19 (CS) is statistically significant and is in positive relationship with proximity (C1, $B$ = 0.177, $\beta$ = 0.223, t = 6.265, ρ<0.000), college image and reputation (C3, $B$ = 0.191, $\beta$ = 0.143, t = 2.281, ρ<0.05), quality education (C7, $B$ = 0.363, $\beta$ = 0.268, t = 4.017, ρ<0.000)) and curriculum delivery (C10, $B$ = 0.186, $\beta$ = 0.141, t = 2.073, ρ<0.05) offered by their ECs. Unstandardized estimates ($B$) revealed that when C1, C3, C7 and C10 is increased by one unit,

**Table 3. Regression analysis CS←(C1 to C11).**

| ECs characteristics (Independent variables, predictors) | Dependable variable—Sustainability under Covid-19 (CS) | | | | | | | | |
|---|---|---|---|---|---|---|---|---|---|
| | Mean (μ) | Unstandardized Coefficients | | Standardized Coefficients | $t$ | $\rho$ | Collinearity Statistics | | Hypothesis Support |
| | | *B* | Std. Error | β | | | Tolerance | VIF | |
| (Constant) | | -0.363 | 0.221 | | -1.644 | 0.101 | | | |
| **C1** | 3.480 | **0.177** | **0.028** | **0.223** | **6.265** | **0.000** | **0.885** | **1.129** | **H2** |
| C2 | 4.120 | -0.035 | 0.069 | -0.027 | -0.503 | 0.615 | 0.379 | 2.637 | H₀2 |
| **C3** | 4.140 | **0.191** | **0.084** | **0.143** | **2.281** | **0.023** | **0.285** | **3.510** | **H2** |
| C4 | 4.060 | 0.008 | 0.083 | 0.006 | 0.094 | 0.925 | 0.285 | 3.505 | H₀2 |
| C5 | 4.020 | -0.025 | 0.070 | -0.021 | -0.361 | 0.719 | 0.346 | 2.889 | H₀2 |
| C6 | 4.070 | -0.030 | 0.075 | -0.023 | -0.401 | 0.688 | 0.337 | 2.970 | H₀2 |
| **C7** | 4.080 | **0.363** | **0.090** | **0.268** | **4.017** | **0.000** | **0.252** | **3.966** | **H2** |
| C8 | 4.020 | 0.037 | 0.073 | 0.028 | 0.502 | 0.616 | 0.372 | 2.690 | H₀2 |
| C9 | 4.080 | 0.050 | 0.082 | 0.036 | 0.601 | 0.548 | 0.314 | 3.185 | H₀2 |
| **C10** | 4.020 | **0.186** | **0.090** | **0.141** | **2.073** | **0.039** | **0.241** | **4.144** | **H2** |
| C11 | 3.870 | 0.091 | 0.056 | 0.081 | 1.638 | 0.102 | 0.463 | 2.158 | H₀2 |

| Regression Model | | | | ANOVAª | | Hypothesis support |
|---|---|---|---|---|---|---|
| *R* | $R^2$ | Adjusted $R^2$ | SE | F-value | *p* | |
| 0.659ª | 0.435 | 0.422 | 0.797 | 35.220 | 0.000ᵇ | *H2* |

Source: Regression analysis run through SPSS.

Note: Values in bold text are significant at $\rho < 0.05$; a: dependent variable (CS), b: predictors (C1 to C11).

CS increases by 0.177, 0.191, 0.363, and 0.186 units, respectively, while keeping other variables stable. On the other hand, C2, C4, C5, C6, C8, C9 and C11 had no statistical relationship with CS in terms of the contribution made. The hypothesis H2 was retained in cases where the independent variables C1, C3, C7 and C10 were statistically related to the dependent variable (CS). On the other hand, the alternative hypothesis (H₀2) was accepted for C2, C4, C5, C6, C8, C9 and C11, where a statistical association did not exist with CS.

## Relationship of sustainability and ECs characteristics within and across students' characteristics (H3)

To test the relationship between ECs characteristics (C1 to C11) and sustainability under the COVID-19 pandemic (CS), i.e. CS←(C1 to C11) across diversity, ANOVA was conducted to predict the statistical significant across the students' characteristics (S1 to S8). To find out accountability of C1 to C11 on CS, within the students' characteristics (S1 to S8), data were split based on categorical variables (student characteristics) with the split file option available in SPSS, and then simple regression was performed. The statistical results are presented in Table 4 and are elaborated below.

The model CS←(C1 to C11) has no statistical significance ($p > 0.05$) on gender ($F = 0.670$, $p = 0.413$), resulting similar kind of opinions across the gender. However, within the groups, for males ($R = 0.66$, $R^2 = 0.44$, $F = 23.961$, $p = 0.000$) and females ($R = 0.70$, $R^2 = 0.49$, $F = 13.394$, $p = 0.000$), it fitted in a similar way with moderate strength of association. CS was significantly predicted by C1, C3, C7 and C10 for males, while C1 and C4 were the significant estimates for females.

**Table 4. Regression analysis CS←(C1 to C11) within students' characteristics.**

| | | ECs Characteristics | | | | | | | | | | |
|---|---|---|---|---|---|---|---|---|---|---|---|---|
| | | C1 | C2 | C3 | C4 | C5 | C6 | C7 | C8 | C9 | C10 | C11 |
| **Gender** | | | | | | | | | | | | |
| Male | | | | | | | | | | | | |
| | μ | 3.48 | 4.10 | 4.13 | 4.03 | 3.95 | 4.04 | 4.07 | 3.98 | 4.03 | 3.99 | 3.82 |
| | β | 0.22 | 0.01 | 0.20 | -0.09 | -0.06 | -0.01 | 0.28 | 0.02 | -0.03 | 0.19 | 0.11 |
| | t | 5.09 | 0.16 | 2.60 | -1.19 | -0.92 | -0.09 | 3.49 | 0.35 | -0.35 | 2.26 | 1.76 |
| | p | **0.00** | 0.88 | **0.01** | 0.24 | 0.36 | 0.93 | **0.00** | 0.73 | 0.73 | **0.02** | 0.08 |
| Female | | | | | | | | | | | | |
| | μ | 3.48 | 4.17 | 4.17 | 4.14 | 4.15 | 4.11 | 4.10 | 4.11 | 4.19 | 4.08 | 3.95 |
| | β | 0.22 | -0.10 | 0.00 | 0.28 | 0.01 | -0.11 | 0.16 | 0.13 | 0.11 | 0.15 | 0.03 |
| | t | 3.39 | -0.86 | 0.01 | 2.59 | 0.12 | -1.08 | 1.29 | 1.24 | 1.09 | 1.24 | 0.39 |
| | p | **0.00** | 0.39 | 0.99 | **0.01** | 0.91 | 0.28 | 0.20 | 0.22 | 0.28 | 0.22 | 0.70 |
| **Socioeconomic status (SES)** | | | | | | | | | | | | |
| Higher SES | | | | | | | | | | | | |
| | μ | 3.51 | 4.17 | 4.18 | 4.11 | 4.05 | 4.09 | 4.11 | 4.05 | 4.10 | 4.07 | 3.89 |
| | β | 0.25 | -0.02 | 0.21 | -0.11 | -0.03 | -0.01 | 0.21 | 0.09 | 0.02 | 0.14 | 0.07 |
| | t | 5.91 | -0.32 | 2.78 | -1.44 | -0.49 | -0.18 | 2.86 | 1.28 | 0.31 | 1.92 | 1.28 |
| | p | **0.00** | 0.75 | **0.01** | 0.15 | 0.62 | 0.86 | **0.00** | 0.20 | 0.76 | 0.06 | 0.20 |
| Lower SES | | | | | | | | | | | | |
| | μ | 3.39 | 3.98 | 4.03 | 3.92 | 3.91 | 3.98 | 3.98 | 3.95 | 4.02 | 3.87 | 3.79 |
| | β | 0.16 | 0.03 | -0.03 | 0.26 | -0.03 | -0.05 | 0.54 | -0.13 | -0.01 | 0.18 | 0.03 |
| | t | 2.61 | 0.39 | -0.25 | 2.22 | -0.26 | -0.51 | 3.32 | -1.30 | -0.09 | 1.10 | 0.25 |
| | p | **0.01** | 0.70 | 0.80 | **0.03** | 0.80 | 0.61 | **0.00** | 0.20 | 0.93 | 0.28 | 0.81 |
| **Native Place** | | | | | | | | | | | | |
| Urban | | | | | | | | | | | | |
| | μ | 3.69 | 4.17 | 4.19 | 4.11 | 4.08 | 4.08 | 4.12 | 4.08 | 4.14 | 4.07 | 3.88 |
| | β | 0.16 | 0.02 | 0.01 | 0.13 | -0.06 | -0.27 | 0.32 | 0.20 | 0.21 | 0.12 | 0.02 |
| | t | 2.64 | 0.20 | 0.07 | 1.35 | -0.71 | -2.61 | 3.47 | 1.93 | 2.14 | 1.21 | 0.34 |
| | p | **0.01** | 0.84 | 0.95 | 0.18 | 0.48 | **0.01** | **0.00** | 0.06 | **0.03** | 0.23 | 0.74 |
| Rural | | | | | | | | | | | | |
| | μ | 3.35 | 4.10 | 4.11 | 4.04 | 3.98 | 4.06 | 4.06 | 3.99 | 4.05 | 3.99 | 3.86 |
| | β | 0.25 | -0.04 | 0.15 | 0.01 | 0.02 | 0.03 | 0.34 | -0.08 | -0.08 | 0.11 | 0.14 |
| | t | 5.58 | -0.67 | 1.83 | 0.09 | 0.24 | 0.40 | 3.30 | -1.09 | -0.96 | 1.20 | 2.03 |
| | p | **0.00** | 0.50 | 0.07 | 0.93 | 0.81 | 0.69 | **0.00** | 0.28 | 0.34 | 0.23 | **0.04** |
| **Pre-college performance** | | | | | | | | | | | | |
| High Scorer | | | | | | | | | | | | |
| | μ | 3.66 | 4.22 | 4.18 | 4.10 | 4.05 | 4.06 | 4.08 | 4.08 | 4.10 | 4.05 | 3.90 |
| | β | 0.34 | -0.18 | 0.17 | -0.10 | 0.01 | 0.07 | 0.30 | -0.09 | 0.00 | 0.35 | -0.03 |
| | t | 6.04 | -2.21 | 1.69 | -0.92 | 0.06 | 0.72 | 2.80 | -0.87 | 0.03 | 3.29 | -0.40 |
| | p | **0.00** | **0.03** | 0.09 | 0.36 | 0.95 | 0.47 | **0.01** | 0.39 | 0.98 | **0.00** | 0.69 |
| Low Scorer | | | | | | | | | | | | |
| | μ | 3.62 | 3.36 | 4.05 | 4.11 | 4.04 | 4.00 | 4.07 | 4.08 | 3.98 | 4.08 | 4.00 |
| | β | 0.16 | 0.07 | 0.12 | 0.06 | 0.00 | -0.10 | 0.25 | 0.06 | 0.07 | 0.04 | 0.14 |
| | t | 3.51 | 0.95 | 1.42 | 0.71 | 0.05 | -1.31 | 2.90 | 0.89 | 0.79 | 0.47 | 2.18 |
| | p | **0.00** | 0.34 | 0.16 | 0.48 | 0.96 | 0.19 | **0.00** | 0.38 | 0.43 | 0.64 | **0.03** |
| **Major Choice** | | | | | | | | | | | | |

*(Continued)*

**Table 4.** (Continued)

| | | ECs Characteristics | | | | | | | | | | |
|---|---|---|---|---|---|---|---|---|---|---|---|---|
| | | **C1** | **C2** | **C3** | **C4** | **C5** | **C6** | **C7** | **C8** | **C9** | **C10** | **C11** |
| Computer allied | | | | | | | | | | | | |
| | μ | 3.54 | 4.16 | 4.15 | 4.09 | 4.05 | 4.09 | 4.12 | 4.07 | 4.12 | 4.07 | 3.95 |
| | β | 0.25 | -0.16 | 0.17 | -0.10 | 0.01 | -0.15 | 0.25 | 0.12 | 0.17 | 0.07 | 0.21 |
| | t | 6.00 | -2.43 | 2.37 | -1.36 | 0.17 | -1.83 | 3.23 | 1.80 | 2.28 | 0.81 | 3.20 |
| | p | **0.00** | **0.02** | **0.02** | 0.17 | 0.86 | 0.07 | **0.00** | 0.07 | **0.02** | 0.42 | **0.00** |
| Non-Computer allied | | | | | | | | | | | | |
| | μ | 3.28 | 4.00 | 4.11 | 3.98 | 3.91 | 4.01 | 3.98 | 3.88 | 3.98 | 3.88 | 3.63 |
| | β | 0.25 | 0.26 | 0.04 | 0.15 | 0.02 | 0.03 | 0.38 | -0.16 | -0.22 | 0.25 | -0.03 |
| | t | 3.59 | 2.92 | 0.35 | 1.21 | 0.18 | 0.34 | 2.65 | -1.84 | -2.30 | 2.09 | -0.37 |
| | p | **0.00** | **0.00** | 0.73 | 0.23 | 0.85 | 0.73 | **0.01** | 0.07 | **0.02** | **0.04** | 0.71 |
| **Delivery mode** | | | | | | | | | | | | |
| Online | | | | | | | | | | | | |
| | μ | 3.19 | 3.20 | 3.98 | 3.97 | 3.97 | 3.87 | 3.94 | 3.95 | 3.86 | 3.95 | 3.92 |
| | β | 0.35 | -0.01 | 0.07 | -0.05 | 0.04 | 0.02 | -0.09 | 0.20 | -0.04 | 0.15 | 0.29 |
| | t | 4.91 | -0.09 | 0.44 | -0.36 | 0.26 | 0.17 | -0.55 | 1.55 | -0.26 | 0.93 | 2.75 |
| | p | **0.00** | 0.93 | 0.66 | 0.72 | 0.80 | 0.87 | 0.58 | 0.12 | 0.80 | 0.36 | **0.01** |
| Hybrid | | | | | | | | | | | | |
| | μ | 3.58 | 4.04 | 4.12 | 3.88 | 3.87 | 3.89 | 3.94 | 3.90 | 3.92 | 3.85 | 3.80 |
| | β | 0.21 | -0.21 | 0.05 | 0.22 | -0.13 | 0.16 | 0.53 | -0.21 | 0.01 | 0.40 | -0.27 |
| | t | 2.58 | -1.47 | 0.36 | 1.55 | -1.01 | 1.19 | 3.73 | -1.57 | 0.08 | 3.03 | -2.16 |
| | p | **0.01** | 0.15 | 0.72 | 0.12 | 0.31 | 0.24 | **0.00** | 0.12 | 0.94 | **0.00** | **0.03** |
| Onsite | | | | | | | | | | | | |
| | μ | 3.57 | 4.22 | 4.23 | 4.18 | 4.14 | 4.19 | 4.20 | 4.15 | 4.21 | 4.13 | 3.96 |
| | β | 0.16 | 0.01 | 0.22 | -0.02 | 0.05 | -0.10 | 0.33 | 0.07 | 0.00 | 0.11 | 0.04 |
| | t | 3.27 | 0.07 | 2.61 | -0.27 | 0.68 | -1.15 | 3.40 | 0.98 | -0.04 | 1.12 | 0.58 |
| | p | **0.00** | 0.94 | **0.01** | 0.79 | 0.50 | 0.25 | **0.00** | 0.33 | 0.97 | 0.27 | 0.56 |
| **Information Source** | | | | | | | | | | | | |
| Non-interactive-offline | | | | | | | | | | | | |
| | μ | 3.41 | 4.10 | 4.06 | 3.98 | 4.02 | 4.05 | 3.98 | 4.01 | 4.05 | 3.93 | 3.67 |
| | β | 0.26 | -0.07 | 0.20 | 0.10 | 0.12 | -0.28 | 0.48 | 0.22 | 0.01 | -0.12 | -0.12 |
| | t | 2.65 | -0.39 | 1.02 | 0.50 | 0.57 | -1.36 | 2.90 | 1.26 | 0.05 | -0.61 | -1.04 |
| | p | **0.01** | 0.70 | 0.31 | 0.62 | 0.57 | 0.18 | **0.01** | 0.21 | 0.96 | 0.54 | 0.30 |
| Non-interactive-online | | | | | | | | | | | | |
| | μ | 3.67 | 4.15 | 4.20 | 4.16 | 4.10 | 4.14 | 4.22 | 4.12 | 4.17 | 4.18 | 3.92 |
| | β | 0.18 | 0.31 | 0.15 | -0.27 | -0.01 | 0.04 | 0.26 | -0.08 | 0.09 | -0.02 | 0.21 |
| | t | 2.86 | 2.98 | 1.45 | -2.48 | -0.09 | 0.33 | 1.80 | -0.88 | 0.84 | -0.18 | 2.62 |
| | p | **0.01** | **0.00** | 0.15 | **0.01** | 0.93 | 0.75 | 0.07 | 0.38 | 0.40 | 0.86 | **0.01** |
| Interactive-offline | | | | | | | | | | | | |
| | μ | 3.68 | 4.21 | 4.18 | 4.08 | 4.04 | 4.18 | 4.07 | 4.03 | 4.12 | 3.99 | 3.95 |

(*Continued*)

**Table 4.** (Continued)

| | | ECs Characteristics | | | | | | | | | | |
|---|---|---|---|---|---|---|---|---|---|---|---|---|
| | | **C1** | **C2** | **C3** | **C4** | **C5** | **C6** | **C7** | **C8** | **C9** | **C10** | **C11** |
| | β | 0.15 | -0.08 | 0.18 | 0.19 | 0.00 | -0.06 | 0.16 | 0.01 | -0.03 | 0.49 | -0.17 |
| | t | 2.04 | -0.83 | 1.11 | 1.07 | 0.01 | -0.50 | 0.95 | 0.09 | -0.28 | 2.79 | -1.24 |
| | p | **0.04** | 0.41 | 0.27 | 0.29 | 0.99 | 0.62 | 0.34 | 0.93 | 0.78 | **0.01** | 0.22 |
| Interactive-online | | | | | | | | | | | | |
| | μ | 3.18 | 4.04 | 4.09 | 4.00 | 3.91 | 3.91 | 4.01 | 3.92 | 3.99 | 3.93 | 3.86 |
| | β | 0.27 | -0.12 | -0.01 | 0.10 | -0.09 | -0.01 | 0.39 | -0.06 | -0.04 | 0.15 | 0.22 |
| | t | 3.93 | -1.01 | -0.10 | 0.97 | -0.80 | -0.06 | 3.04 | -0.54 | -0.32 | 1.07 | 2.11 |
| | p | **0.00** | 0.32 | 0.92 | 0.33 | 0.43 | 0.95 | **0.00** | 0.59 | 0.75 | 0.29 | **0.04** |
| **Human Influence** | | | | | | | | | | | | |
| College | | | | | | | | | | | | |
| | μ | 3.70 | 4.18 | 4.14 | 4.09 | 4.04 | 4.09 | 4.14 | 4.04 | 4.12 | 4.03 | 3.89 |
| | β | 0.18 | 0.00 | 0.19 | -0.07 | -0.03 | -0.05 | 0.27 | 0.20 | -0.10 | 0.10 | 0.19 |
| | t | 3.22 | 0.06 | 2.20 | -0.79 | -0.33 | -0.70 | 3.02 | 2.53 | -1.14 | 0.99 | 2.59 |
| | p | **0.00** | 0.95 | **0.03** | 0.43 | 0.75 | 0.48 | **0.00** | **0.01** | 0.26 | 0.32 | **0.01** |
| Community | | | | | | | | | | | | |
| | μ | 3.13 | 4.07 | 4.15 | 3.99 | 3.97 | 4.04 | 3.99 | 3.96 | 4.01 | 3.97 | 3.78 |
| | β | 0.27 | -0.16 | 0.11 | 0.05 | -0.11 | 0.07 | 0.20 | -0.19 | 0.17 | 0.44 | -0.05 |
| | t | 3.99 | -1.45 | 0.81 | 0.37 | -1.10 | 0.55 | 1.51 | -1.68 | 1.35 | 3.10 | -0.59 |
| | p | **0.00** | 0.15 | 0.42 | 0.72 | 0.27 | 0.58 | 0.13 | 0.10 | 0.18 | **0.00** | 0.55 |
| Family | | | | | | | | | | | | |
| | μ | 3.63 | 4.16 | 4.20 | 4.16 | 4.10 | 4.11 | 4.12 | 4.03 | 4.12 | 4.09 | 3.94 |
| | β | 0.36 | 0.11 | -0.16 | 0.13 | 0.31 | -0.36 | 0.55 | -0.09 | 0.20 | -0.17 | -0.04 |
| | t | 3.84 | 0.68 | -0.83 | 0.70 | 1.45 | -1.77 | 2.01 | -0.57 | 1.04 | -1.07 | -0.25 |
| | p | **0.00** | 0.50 | 0.41 | 0.48 | 0.15 | 0.08 | **0.05** | 0.57 | 0.30 | 0.29 | 0.80 |
| School | | | | | | | | | | | | |
| | μ | 3.25 | 3.96 | 4.04 | 4.00 | 3.92 | 3.94 | 4.02 | 4.10 | 4.06 | 4.00 | 3.86 |
| | β | 0.19 | -0.44 | 0.27 | -0.24 | 0.05 | 0.47 | 0.35 | 0.07 | 0.20 | -0.35 | 0.34 |
| | t | 1.66 | -1.13 | 0.68 | -0.77 | 0.19 | 1.47 | 0.82 | 0.17 | 0.91 | -1.09 | 1.87 |
| | p | 0.11 | 0.27 | 0.50 | 0.45 | 0.85 | 0.15 | 0.42 | 0.87 | 0.37 | 0.28 | 0.07 |

| Students Characteristics | R | $R^2$ | Adjusted $R^2$ | SE | F-value | p | Hypothesis support (across) | Hypothesis support (within) |
|---|---|---|---|---|---|---|---|---|
| **Gender** | | | | | 0.670 | 0.413 | $H_03$ | |
| Male | 0.66 | 0.44 | 0.42 | 0.85 | 23.961 | 0.000 | | **H3** |
| Female | 0.7 | 0.49 | 0.45 | 0.66 | 13.394 | 0.000 | | **H3** |
| **Social Class** | | | | | 3.678 | 0.056 | $H_03$ | |
| Higher social class | 0.81 | 0.66 | 0.63 | 0.62 | 19.632 | 0.000 | | **H3** |
| Lower social class | 0.62 | 0.39 | 0.37 | 0.84 | 21.899 | 0.000 | | **H3** |
| **Native Place** | | | | | 3.393 | 0.066 | $H_03$ | |
| Urban | 0.71 | 0.5 | 0.47 | 0.74 | 16.553 | 0.000 | | **H3** |
| Rural | 0.66 | 0.43 | 0.41 | 0.82 | 21.569 | 0.000 | | **H3** |
| **Pre-college performance** | | | | | 1.474 | 0.225 | $H_03$ | |
| High Scorer | 0.67 | 0.45 | 0.42 | 0.77 | 14.913 | 0.000 | | **H3** |
| Low Scorer | 0.68 | 0.46 | 0.44 | 0.81 | 22.443 | 0.000 | | **H3** |
| **Major Choice** | | | | | **5.965** | **0.015** | **H3** | |
| Computer allied | 0.65 | 0.42 | 0.41 | 0.79 | 24.982 | 0.000 | | **H3** |

(*Continued*)

**Table 4.** (Continued)

| | | C1 | C2 | C3 | C4 | C5 | C6 | C7 | C8 | C9 | C10 | C11 |
|---|---|---|---|---|---|---|---|---|---|---|---|---|
| | | | | | | | **ECs Characteristics** | | | | | |
| Non- Computer allied | | | 0.77 | 0.6 | 0.56 | 0.73 | 15.921 | 0.000 | | | H3 | |
| **Delivery mode** | | | | | | | | 7.756 | **0.000** | H3 | | |
| Online | | | 0.7 | 0.49 | 0.45 | 0.87 | 10.642 | 0.000 | | | H3 | |
| Hybrid | | | 0.7 | 0.49 | 0.43 | 0.7 | 8.267 | 0.000 | | | H3 | |
| Onsite | | | 0.66 | 0.44 | 0.42 | 0.72 | 19.011 | 0.000 | | | H3 | |
| **Information Sources** | | | | | | | | 4.036 | **0.007** | H3 | | |
| Non-interactive-offline | | | 0.66 | 0.44 | 0.35 | 0.84 | 4.868 | 0.000 | | | H3 | |
| Non-interactive-online | | | 0.7 | 0.49 | 0.45 | 0.73 | 13.126 | 0.000 | | | H3 | |
| Interactive-offline | | | 0.77 | 0.59 | 0.55 | 0.82 | 13.422 | 0.000 | | | H3 | |
| Interactive-online | | | 0.63 | 0.4 | 0.35 | 0.78 | 8.744 | 0.000 | | | H3 | |
| **Human Influence** | | | | | | | | 3.823 | **0.047** | H3 | | |
| College | | | 0.71 | 0.5 | 0.47 | 0.78 | 19.333 | 0.000 | | | H3 | |
| Community | | | 0.65 | 0.42 | 0.37 | 0.85 | 9.052 | 0.000 | | | H3 | |
| Family | | | 0.72 | 0.51 | 0.44 | 0.67 | 7.451 | 0.000 | | | H3 | |
| School | | | 0.74 | 0.55 | 0.43 | 0.82 | 4.39 | 0.000 | | | H3 | |

Source: Regression analysis run through SPSS

Note: Values in bold are significant at $p<0.05$. Dependent variable is CS, predictors are C1 to C11

In the case of the SES students, the model CS←(C1 to C11) did not show any statistical significance ($F = 3.678$, $p = 0.056$) as $p>0.05$. However, it is better suited to higher SES ($R = 0.81$, $R^2 = 0.66$, $F = 19.632$, $p = 0.000$) with substantial strength of association than lower SES ($R = 0.62$, $R^2 = 0.39$, $F = 21.899$, $p = 0.000$) with low strength of association, indicating significance within the groups. C1, C3 and C7 for the higher SES and C1, C4 and C7 for the lower SES contributed significantly in predicting CS, within the groups.

For the group native place ($F = 3.393$, $p = 0.066$), the model CS←(C1 to C11) was statistically insignificant with $p>0.05$. The strength of association CS←(C1 to C11) for both; urban and rural, within the group was significant and moderate. Within the groups, CS was contributed by the effect of C1, C6, C7 and C9 for urban students ($R = 0.71$, $R^2 = 0.50$, $F = 16.553$, $p = 0.000$) and C1, C7 and C11 for rural students ($R = 0.66$, $R^2 = 0.43$, $F = 21.569$, $p = 0.000$).

There were no disparities noted across the group of pre-college performance ($F = 1.474$, $p = 0.225$), for the model CS←(C1 to C11) with $p>0.05$. However, within the group, CS was significantly contributed by C1, C2, C7 and C10 for high scorers ($R = 0.67$, $R^2 = 0.45$, $F = 14.913$, $p = 0.000$) with substantial strength of association, and C1, C7 and C11 for low scorers ($R = 0.68$, $R^2 = 0.46$, $F = 22.443$, $p = 0.000$) with moderate strength of association.

In the case of major choice ($F = 5.95$, $p = 0.015$), the regression model CS←(C1 to C11) showed statistical significance, indicating dissimilarities across the groups. The model is significant within the both groups but more appropriate for students adopting non-computer allied majors ($R = 0.77$, $R^2 = 0.60$, $F = 15.921$, $p = 0.000$) with substantial strength of association than for students accepting computer allied majors ($R = 0.65$, $R^2 = 0.42$, $F = 24.982$, $p = 0.000$) with moderate strength of association. C1, C2, C3, C7, C9 and C11 and C1, C2, C7, C9 and C10 were the key significant influencers within the groups, in predicting CS for students adopting computer allied majors and non-computer allied majors, respectively.

Understanding delivery mode ($F = 7.756$, $p = 0.000$), it exhibited statistical significance for the regression model CS←(C1 to C11) across the groups displaying differences. It had more

bearing on online ($R = 0.70$, $R^2 = 0.49$, $F = 10.642$, $p = 0.000$) and hybrid ($R = 0.70$, $R^2 = 0.49$, $F = 8.267$, $p = 0.000$) than onsite mode ($R = 0.66$, $R^2 = 0.44$, $F = 19.011$, $p = 0.000$), with moderate strength of association for all. Only two significant predictors, C1 and C11 within the group, governed CS in the case of the online mode. CS as estimated by virtue of C1, C7 C10 and C11 and C1, C3 and C7 for hybrid and onsite delivery modes, respectively, within the groups.

The regression model CS←(C1 to C11) for information source ($F = 4.036$, $p = 0.007$) across its groups had shown statistical significance as $p < 0.05$ showing dissimilarities. The model outfitted better for the non-interacting and online source (website) ($R = 0.77$, $R^2 = 0.59$, $F = 13.126$, $p = 0.000$), and Interactive-offline (face-to-face counselling) ($R = 0.70$, $R^2 = 0.49$, $F = 13.422$, $p = 0.000$), then non-interactive (print and media advertisement) ($R = 0.66$, $R^2 = 0.44$, $F = 4.868$, $p = 0.000$) and Interactive-online sources (social media) ($R = 0.63$, $R^2 = 0.40$, $F = 8.744$, $p = 0.000$) with moderate to substantial strength of association. Within the groups, CS was significantly contributed by C1, C2, C4 and C11 for Non-interactive-online, C1 and C7 for Non-interactive-offline, C1 and C10 for Interactive-offline, while C1, C7, C11 for Interactive-online.

In view of human influence ($F = 3.823$, $p = 0.047$), the regression model with $p > 0.05$, was statistically significant presenting differences across the groups. The model is better poised for school ($R = 0.74$, $R^2 = 0.55$, $F = 4.390$, $p = 0.000$) with substantial strength of association, for family ($R = 0.72$, $R^2 = 0.51$, $F = 7.451$, $p = 0.000$), for college representative ($R = 0.71$, $R^2 = 0.50$, $F = 19.333$, $p = 0.000$) than community ($R = 0.65$, $R^2 = 0.42$, $F = 9.052$, $p = 0.000$) with moderate strength of association. Within the groups, CS was significantly contributed by C1, C3, C7, C8 and C11 for college representatives, C1 and C10 for community, and C1 and C7 for families. No college characteristics (C1 to C11) are significantly accounted for CS when school is an influencing element.

There was a statistically significant relationship originated in predicting the dependent variable (CS, continuous variable) on account for by independent variables, C1 to C11 (continuous variables) across diversity i.e. students' characteristic S5 to S8 (categorical variables), supporting to hypothesis H3 (Refer Table 4). However, it was statistically insignificant across students' characteristic S1 to S4, displaying similar attitudes on CS←(C1 to C11), hence, alternative hypothesis $H_03$ was accepted.

Next, within all groups (S1 to S8) it discovered a statistical significance over sustainability of ECs (CS) contributed by ECs characteristics (C1 to C11) i.e. the regression model CS←(C1 to C11) statistically significant within all students characteristics

## Statistical analysis, assessment of outcomes and interpretations

**ECs characteristics influencing students' choice for ECs.** Choice for ECs was significantly influenced by a number of in-built characteristics of ECs, and this research confirmed the significance of those. There were twelve defining characteristics of a ECs; eleven were the same as they've always been before pandemic and one is new; sustainability under COVID-19, which has been endorsed in light of the current pandemic. While having similar functionality in both the pre-pandemic and COVID-19 pandemic settings, ECs characteristics proved to be a crucial in deciding selecting ECs for students.

Curriculum delivery ($\mu = 4.020$, $R^2 = 0.761$) [99,131], quality education ($\mu = 4.080$, $R^2 = 0.756$) [14,208–210], image and reputation ($\mu = 4.140$, $R^2 = 0.718$) [12,185,346–348], faculty ($\mu = 4.060$, $R^2 = 0.715$) [184,185,191–193], safe and secured campus ($\mu = 4.080$, $R^2 = 0.686$) [57,311], campus placement activities ($\mu = 4.070$, R2 = 0.663) [205], alumni ($\mu = 4.020$, $R^2 = 0.654$) [123,200,201], infrastructure and facilities ($\mu = 4.020$, $R^2 = 0.628$) [349], and location

and locality ($\mu$ = 4.120, $R^2$ = 0.621) [179,181,350] were the powerful influencing ECs characteristics of highest importance in making their EC's choice for the overall diversity under study that firmly supported previous studies. It appeared that students disregarded the issues of proximity, value for money and sustainability while picking an EC in a COVID-19 situation.

Sustainability ($\mu$ = 3.640, $R^2$ = 0.435) though appeared of lower importance, but proved to be reliable character in selecting ECs, which in accordance to the study of [351], which states that it is the subject of matter to prospective students as well as the public during enrollments. Similar to the findings of [352] sustainability under COVID-19 ($\mu$ = 3.640, $R^2$ = 0.435) has revealed to be of less importance.

Like to the study of [296], this investigation discovered that proximity ($\mu$ = 3.480, $R^2$ = 0.297) i.e. geographical closeness had an impact on EC selection but with lower priority. This constructive approach perhaps explain the younger generation's desire to follow their lifelong aim of becoming engineers and may indicate positive sign for ECs located at a far distance. As number of ECs are less to accommodate the aspiring population willing to join EE, and are scattered at a far distance from their hometowns. But, prospective students must persevere in order to accomplish their engineering passion at a distant location. Other key benefits associated with the premium programme such as engineering, cannot be ignored due to proximity issues. Proximity and opportunities associated with the ECs are frequently embodied as two sides of a coin, from which students must choose one. This perhaps are the reasons why the students from all backgrounds have acknowledged lower importance to proximity associated with distance, time and cost, than other ECs characteristics controlling decisions about ECs choice.

Lowered emphasis given to value for money ($\mu$ = 3.870, $R^2$ = 0.539) might indicate that spending money is not difficult when considering its associated benefits, as similarly shown by the research of [297].

In response to research question RQ1, all traditional ECs characteristics, including newly emerged sustainability of ECs under COVID-19 pandemic have derived variable scores on ranking of importance, but have been found dominant in dominating ECs choice decisions under COVID-19 pandemic, as specified by various studies like [189,194] belonging to different cultures on a regular basis.

## Insights of diversity on sustainability of ECs

Students' perceptions on the sustainability of ECs under COVID-19 were insignificant across their characteristics; demographic: gender (S1), socioeconomic: SES (S2), geographical: native place (S3), and academic: pre-college performance (S4). This is due to the fact that the pandemic has affected the education of all students [353], whether they are male or female, from a higher or lower SES background, live in urban or rural areas, or be high or low academic scores. The findings confirm the prior studies of [354,355] in terms of pandemic impact on female students and academic performance demonstrating no intersectional stigma. Nevertheless, these findings vary from prior research [294] that found a strong pandemic effect on SES and academic performance [356], gender [355], and geographical traits [79]. Students' perceptions on psychological and behavioural traits were significantly influenced by sustainability ECs under COVID-19.

Major choices (S5, $F$-value = 8.333, $p$ = 0.004), did have statistically significant impact ($p < 0.05$) on sustainability under COVID-19, with the students enrolling in computer allied courses ($\mu$ = 3.72) placing a higher priority on sustainability than the students with non-computer related majors ($\mu$ = 3.42). Until now, in India, delivery of HE during pandemic situations has been performed online with students sitting and performing at home with their on line

sources like computer/laptop/mobile that are effective in HE delivery [297]. Subjects covered in computer allied majors can be learned and understood at home with a hi-tech online platform. This insight has made computer allied majors a greater sustainability under COVID-19. Till the execution of this survey, most EE delivery in India has been done online, with students sitting and performing at home with their on-line sources such as computer/laptop/mobile effective in receiving online delivery [297]. Computer associated key subjects can be learnt and comprehended at home using a high-tech online platform. This realisation may has increased the applicability of these majors under pandemic for the students who enrolled in it which in accordance to the study of [310,357]. Non-computer associated majors, on the other hand, are difficult to acquire and grasp from home [358], unsuitable for online learning [244,359] and might cause significant academic loss [277] during the epidemic. As a result, students with non-computer associated majors estimated lesser sustainability with online delivery, which is consistent with the findings of [280].

Course delivery (S6, $F$-value = 25.601, $p$ = 0.000), had significant influence on sustainability making a greater impression on the students' intentions about onsite (μ = 3.92) and hybrid (μ = 3.48) than online delivery (μ = 3.19). It contributes to the study of [280] in a global setting in a way that onsite mode is effective in teacher-student interactions thereby supporting sustainability. With the reopening of ECs during the pandemic, the hybrid delivery method, a combination of online theoretical pedagogy and onsite pedagogy for lab work, with safe distancing measures, proved to be successful [107]. Students are interested in onsite pedagogy since it allows them to learn and grasp difficult subjects in person. As a result, engineering students place a higher value on onsite and hybrid pedagogy. This is conceivably the reason why greater sustainability is convinced by the students preferring onsite pedagogy than the other modes that is identical with the previous studies of [360–362] connected to EE. The study's findings about the lesser sustainability professed by the students who prefer to opt online mode are consistent with earlier studies that indicated online mode of teaching to be challenging to teach [358], unfavourable for learning from home [244,359], leading to academic loss [277], ineffectiveness [363], creates a skills gap [290], psychologically challenging [364] and digital difficulties [270] during pandemics, during pandemics, however are different from the study of [365] indicating its appropriateness in EE. Another Indian study [273] noting no disparities in between online and onsite learning due the impact of pandemic is contradictory to this study.

The sustainability (S7, $F$-value = 2.686, $p$ = 0.046) under COVID-19 pandemic is significantly associated ($p<0.05$) with human influence. Family members (μ = 3.84) as a close credible source of advice [366] and college representatives (μ = 3.69), as a direct source [367], as a significant influencer in guiding and providing superior information under pandemic conditions, have encouraged students in EC choice decisions hence professed greater sustainability. It is consistent with the previous study of [119], which indicated the importance of parents and college stakeholders in influencing college choice decisions. Students who are influenced by their community (μ = 3.52), such as friends and relatives, may be unable to be influenced due to a lack of information about current provisions and happenings of ECs during a pandemic situation, and thus place a lower value on sustainability.

Students' psychological perspectives on information sources (S8, $F$-value = 2.560, $p$ = 0.049) used in making EC selection had a significant relationship with EC sustainability under COVID-19. Interactive-offline sources (μ = 3.71) such as face-to-face communications, and non-interactive-online information sources (μ = 3.77) such as websites, had a greater impact on proposing ECs sustainability due to the possibility that they had provided reliable, detailed, and appropriate information to support their decision-making. This supports previous research findings that revealing websites [368–370], and face-to-face communications [371,372] are important for facilitating college selection decisions. Non-interactive offline

sources such as print materials, advertising media, and outdoor activities, as well as interactive online sources such as social media, had less of an impact on students' intention about sustainability of ECs under COVID-19. The most likely reason is that the ECs and the community in the COVID-19 situation might have unable them to communicate properly and frequently on their social media networks due to the panic environment and other priority works related to pandemic measures. Next ECs may not have been able to distribute print materials due to pandemic restrictions. Lesser sustainability reported by this study's students who used social media ($\mu$ = 3.46) as their primary source of information in making EC choice contradicts previous studies of [158,370,373] that identified social media as a leading, rich, and interactive communication source, as well as the study of that reveals it as an effective tool in decision making [374].

In general, explanation to research question RQ1 is that the perceived intentions as per the student's demographic, socioeconomic, geographic, and pre-college academic characteristics suggested no influence on sustainability ECs under pandemic. When another student's trait - psychological and behavioral characteristics, such as choice for major, course delivery, human influence, and information sources are considered, they perceived significantly in a different way.

## Effect of ECs characteristics on sustainability

The regression model CS←(C1 to C11) ($R$ = 0.659, $R^2$ = 0.435, $F$ = 35.220, $p$ = 0.000) presented in Table 3, was statistically significant ($\rho$<0.05) in predicting the sustainability of EC under COVID-19 by virtue of eleven characteristics affiliated to ECs. The statistical results revealed that characteristics of ECs such as proximity, image and reputation, quality education and curriculum delivery that were significantly effective with moderate to strong accountability in predicting the sustainability of ECs under the COVID-19 pandemic. Other EC characteristics, such as location and locality, faculty, alumni, campus placements, infrastructure and facilities, safety and security and value for money were found statistically insignificant in estimating the sustainability of EC under the COVID-19 pandemic.

This study has indicated that as proximity (C1, B = 0.177, $\beta$ = 0.223, $t$ = 6.265, $p$ = 0.000) improves, i.e. nearness of EC to the hometown enhances, sustainability of ECs increases. This could be because EC's proximity to their hometown reduces the distance travelled, saves time and money for the family [11], and maintains students' health-related safety and security, making it favorable to sustain. More importantly, it sustains the risk of coronavirus infection due to less travel. [173]. Based on sustainability it also creates favorable conditions for ECs located near students' markets, giving ECs first priority for local students when making EC selection [375], especially in a pandemic situation.

This study discovered that the greater the perceived image and reputation of an EC (C3, B = 0.191, $\beta$ = 0.143, $t$ = 2.281, $p$ = 0.023), the greater the perception of that EC's sustainability under COVID-19. The key dimensions of image and reputation that will be spawned during a crisis situation [258] are word of mouth [186] and trust and beliefs [376]. In conclusion, an EC with a positive image and reputation is more probable to preserve EE under the COVID-19 situation, which increases students' desire for ECs. In this regard, this study supports the study of [377], which states that sustaining a situational crisis is a function of reputation and trust.

This study found that quality education (C7, B = 0.363, $\beta$ = 0.268, $t$ = 4.017, $p$ = 0.000) is a significant predictor of sustainability with a positive connection, indicating that a higher level of quality education leads to a higher level of sustainability during the COVID-19 pandemic. As a result, if the EC provides high-quality educational services, it is perceived to be a good fit under COVID-19, sympathetic to the postulated made by the study of [378].

The most difficult challenge for engineering studies is curriculum delivery during pandemics, and reconsideration of its structure to balancing pandemic influence is an urgent need of the hour [379]. In pandemic situations, successful curriculum delivery requires access flexibility, practice of knowledge, skill building, and keeping students' interests alive by instilling pandemic measures may online or onsite. This viewpoint is supported by this study, which finds that sustainability under COVID-19 is a significantly contributed by curriculum delivery (C10, B = 0.186, β = 0.141, $t$ = 2.073, $p$ = 0.039). Students in engineering are more concerned about curriculum delivery [104] that provides them with the right fairness and justification for becoming competent engineers. As a result, students may have thought of greater sustainability of ECs is due to superior curriculum delivery under the COVID-19 pandemic.

The sustainability of ECs in the face of the COVID-19 pandemic is statistically influenced and contributed by EC characteristics such as proximity, image and reputation, quality education and curriculum delivery, with a positive impact, supporting an earlier study of [352] in the context of Indian EE. When the influencing predictors; proximity, image and reputation, quality education and curriculum delivery increase by 0.177, 0.191, 0.363, and 0.186 units, respectively, the sustainability of ECs increases by one unit while the other predictors kept constant. As discussed above in response to research question RQ2, as proximity, image and reputation, quality education and curriculum delivery of ECs are pragmatic in regulating the sustainability of ECs during a pandemic.

## Effect of ECs characteristics on sustainability within and across diversity

Table 4 provides an explanation of how the characteristics of ECs (C1 to C11) used in the regression model CS←(C1 to C11) to predict the sustainability of ECs under COVID-19 differ within and across student characteristics (S1 to S8)

## Effect of proximity

The study found that proximity (C1, B = 0.177, β = 0.223, $t$ = 6.265, $p$ = 0.000) is a substantial contributor in assessing the sustainability of ECs under COVID-19. This progression has been established effectively within all groups.

The students are rooted in the local environment and unwilling to leave them as they are much familiar with local issues and measures of pandemic. This might be the thought among the students to sustain their local social issues. Therefore the students from all walks of life admitted proximity's accountability for sustainability of ECs. Similar finding were noted by the study of (Mahajan & Patil, 2021) where students of different sexes, economic backgrounds, regions of origin, varied behavior, and choices agree that the closer an EC is more sustainable under COVID-19. This is study acknowledge the findings of [380] that discovered local students were fascinated to the nearby universities during pandemics in regard to sustainability.

## Effect of location and locality

Although the study established that the location and locality (μ = 4.120) had a higher importance in determining which ECs to select, but surprisingly demonstrated that it (C2, B = -0.035, β = -0.027, $t$ = -0.503, $p$ = 0.615) had no impact on the long-term sustainability of ECs in the face of a pandemic across the groups. However, it has created significant influence on sustainability within the groups such as high pre-college performance (high scorers), students with computer-allied majors as well as non-computer allied majors and students influenced by non-iterative-online information source.

The high pre-college performers (β = -0.18, $t$ = -2.21, $p$ = 0.03) and students pursuing computer-allied majors (β = -0.16, $t$ = -2.43, $p$ = 0.02), expressed less than expected sustainability

with greater location and locality. The high performing students and students enrolled in computer allied majors had more concerns about the prestige along with the employment and earnings outcomes associated with the ECs. The state of the labour market [381] and unemployment statistics in the region [179] is a key driver of location, which turned out to be a severe cause in the pandemic with dwindling jobs in area where the ECs are situated. Second, availability of emerging and computer allied majors in ECs decreases the function of their location and further to avail education in prestigious EC, students have to ignore location restrictions [380]. As a result, these students who were concerned about their employment and program specific benefits during the pandemic may have postulated comparatively lower sustainability with the increased location and locality.

On the other hand, students with non-computer allied majors ($\beta = 0.26$, $t = 2.92$, $p = 0.000$) concluded that enhanced location and loyalty increases sustainability. These students may have envisioned a formulaic way taking into account location and locale with its speciousness, airiness, and safety to sustain their lab practical and field work during the epidemic.

Under the Covid-19 scenario, students who trusted websites ($\beta = 0.31$, $t = 2.98$, $p < 0.000$) for seeking accurate information in their EC selection are supposed to sight better sustainability with better location and locality. This might be because websites of ECs are usually displayed with appealing images and videos that convey more specific location narrative [174] creating the basic sense about the location and locality of ECs. According to the study of [368,369], websites are a valuable resource for propagating positive word of mouth, which in turn attracts potential students.

## Effect of image and reputation

Image and reputation (C3, $\beta = 0.143$, $t = 2.281$, $p = 0.023$) was found to be substantially associated to the sustainability of ECs during a pandemic. Within the groups such as male students, students from higher SES, students enrolled in computer allied majors, students opting onsite curriculum delivery and students influenced by ECs' representatives, image and reputation is dominant player in governing sustainability.

It also has a significant influence on male students ($\beta = 0.2$, $t = 2.6$, $p = 0.01$) and students enrolled in computer allied majors ($\beta = 0.170$, $t = 2.37$, $p < 0.05$), who believed that the higher image and reputation associated with ECs and majors provides more sustainability under COVID-19. Similar kind of results were noted by [205,382–384] where male students and students adopting high earning majors and repute were fascinated by ECs carrying respectable image and reputation.

Higher SES confers better esteem, culture, image, and prestige, which explains why students from higher SES ($\beta = 0.21$, $t = 2.78$, $p = 0.01$) and computer allied majors ($\beta = 0.17$, $t = 2.37$, $p = 0.02$) reported greater sustainability under COVID-19 with greater image and reputation of their ECs. Yet, according to [348] and as evidenced by this study, it is ineffective to recruit a group of lower SES strata for engineering programs.

Image and reputation of ECs are influenced by onsite evidences such as tangible facilities, physical infrastructure, face-to-face encounters, and support services [385], which are the substantial assets during pre-pandemic in the delivery of EE. This might be the reason why the students who have opted the onsite option ($\beta = 0.22$, $t = 2.61$, $p = 0.01$) intended higher sustainability of ECs under COVID-19 with high reputation and image arising due to tangible evidences.

Stakeholders of ECs, such as students and staff, were identified to be essential information sources in generating good word-of-mouth, hence enhancing institutions' image and reputation [386]. Furthermore, it is demonstrated that these genuine, optimistic, and trustworthy statements assuring fitness of ECs for continuing and supporting engineering studies during

COVID-19 might perhaps professed positive feelings among the students. As a result, students for whom college representatives ($\beta = 0.19$, $t = 2.20$, $p < 0.05$) were the prime influential for their EC selection decision, had felt stronger sustainability under the COVID-19 pandemic with better image and reputation.

## Effect of faculty

Faculty (C4, $\beta = 0.006$, $t = 0.094$, $p = 0.925$) were discovered to be a non-crucial contributor in building up the sustainability of ECs in the presence of a pandemic. On the other hand, it is significant character of ECs which has accrued sustainability within the groups of female, lower SES, and students preferring non-interactive-online information.

However, females and students from lower SES, desire faculty to influence them [61,387], want them to increase their psychological well-being [388], affinity, confidence, sense of belonging, and satisfaction [141] for supporting their college choice decisions and their inclusiveness [70] during a pandemic. This might be the notion among the females ($\beta = 0.28$, $t = 2.59$, $p < 0.05$) and lower SES ($\beta = 0.26$, $t = 2.22$, $p < 0.05$) that ECs with the best faculty profile had a better sustainability in the face of a pandemic. They might be considering faculty as facilitators and mentors in sustaining their career paths [54,56]. In accordance with the findings of [389], this might be the reason why these students have identified faculty as a long-term influencer for their inclusion in engineering professions.

Based on teacher profile, students who utilized non-interactive-online sources such as websites as their major information source in their EC selection rated less sustainability than expected. This possibly was due to the fact that, faculty information and their involvement on the majority of websites of ECs were either missing or not clearly explaining in depth to reach, producing an information gap [194] for making a decision. As a consequence, students who utilized websites ($\beta = -0.27$, $t = -2.48$, $p < 0.05$) perceived considerably less sustainability due to predictions about the faculty profile.

## Effect of alumni

Finding of this study discovered that although alumni profile ($\mu = 4.020$) appeared on the upper side of priority list of importance vital in EC selection, ECs alumni (C5, $\beta = -0.021$, $t = -0.361$, $p = 0.719$) were not responsible in counting EC's sustainability, within as well as across the diverse groups. This might be because to alumni network being inaccessible, idle, and inconsistent throughout the pandemic to guide and inspire prospective students, or they could be oblivious of EC's most current pandemic incidents that needed to speak about.

## Effect of campus placements

Campus placement (C6, $\beta = -0.023$, $t = -0.401$, $p = 0.688$) did not play a significant role in determining ECs' sustainability across the diversity. Within the group of students residing in urban area it has negative impact on sustainability. The bulk of entry-level jobs in the engineering profession, notably software and IT professions, are available primarily in companies located in highly urbanised cities [390] that accommodate a densely populated population and have been determined to be dangerous by the COVID-19. During the epidemic, employment in these urbanised cities decreased. Students residing in urbanized cities are fully aware of this. As a result, students perchance whose hometown is an urban area ($\beta = -0.27$, $t = -2.61$, $p = 0.01$) felt somewhat less intentions than expected on the sustainability of EC under the COVID-19 scenario with higher campus placement activities of ECs.

### Effect of quality education

Under COVID-19, the educational quality (C7, $\beta$ = 0.268, t = 4.017, p = 0.000) has considerably estimated the ECs' sustainability across the groups. As a result, its impact reflected positively across and within the student groups based on gender (female), SES status (higher and lower), geographical location (urban as well as rural), pre-college performance (high and low scorers), and psychological and behavioral like major choice, curriculum delivery (hybrid and onsite), preferred information sources (non-interactive-offline and interactive-online) and human influence (college representatives and family) expounding greater the quality of education and greater is the experience of sustainability under the COVID-19 pandemic.

The application of quality education in terms of competency, attitude, content, delivery, and reliability during the pandemic is critical [391], with the feasibility of switching attractions to HEIs [392]. The study has disclosed that because of greater quality education, greater sustainability was perceived by higher SES students ($\beta$ = 0.21, $t$ = 2.86, $p$<0.000) as well as lower SES ($\beta$ = 0.54, $t$ = 3.32, $p$<0.000), urban students ($\beta$ = 0.32, $t$ = 3.47, $p$<0.000) along with rural students ($\beta$ = 0.34, $t$ = 3.3, $p$<0.000), high academic performers ($\beta$ = 0.3, $t$ = 2.80, $p$<0.000) in addition to low academic performers ($\beta$ = 0.25, $t$ = 2.90, $p$<0.000), students enrolled in computer allied majors ($\beta$ = 0.25, $t$ = 3.23, $p$<0.000) as well as students enrolled in non-computer allied majors ($\beta$ = 0.38, $t$ = 2.65, $p$<0.000).

The higher intellectual talents and abilities associated with male students make them adhesive to STEM education [70] than their counterparts. This demands more interactive and qualitative pedagogy to sustain their requirements. This might be the motive for male ($\beta$ = 0.28, $t$ = 3.49, $p$<0.000) and for professing better quality education with the better sustainability under the COVID-19 pandemic.

The concept of 'quality' is most appropriate and defensible in the context of EE [102], when education delivery is practical, which is attainable with physical / in-person delivery [106]. For onsite setups, the quality standards [393,394] are critical. This may be the concerns for the students opting for onsite mode ($\beta$ = 0.33, $t$ = 3.40, $p$ = 0.000) and hybrid mode ($\beta$ = 0.53, $t$ = 3.73, $p$ = 0.000) to intend to sustain academic interest during the pandemic.

Students who used non-interactive offline communication ($\beta$ = 0.48, $t$ = 2.90, $p$<0.05), such as print material and media advertisements controlled by colleges, and interactive online sources, such as social media ($\beta$ = 0.39, $t$ = 3.04, $p$<0.000), after exhibiting presentation on quality measures were found to be better reflectors of quality measures. Perhaps this might be the reason why students who used them as primary information sources felt increased sustainability due to higher quality education provided by ECs.

Similarly, students who influenced by family ($\beta$ = 0.55, $t$ = 2.01, $p$ = 0.049) and EC representatives ($\beta$ = 0.27, $t$ = 3.02, $p$ = 0.000), believed that as education quality improves, EC sustainability under COVID-19 improves. This suggests that the exceptional quality preparations made during the COVID-19 epidemic had a higher favorable influence on family and college representatives, who further establish faith and reliance in promoting sustainability of ECs under pandemic conditions.

### Effect of infrastructure and facilities

Despite the fact that it is said to as a vital support system that is rendered to continue learning, this study discovered that infrastructure and facilities (C8, $\beta$ = 0.028, t = 0.502, $p$ = 0.616) was unsuccessful to make a substantial contribution in managing the sustainability of ECs across students' groups during a pandemic, however was significant within the group of students influenced by ECs' representatives.

ECs representatives who are custodians and users of ECs physical and tangible assets enhance their skills in simplifying EE for students. As they are only the one who able to explain its value during a pandemic, students persuaded by ECs representatives (β = 0.20, $t$ = 2.53, $p$ = 0.01) concluded that increased facilities and infrastructure resulted in higher sustainability under COVID-19.

## Effect of safety and security

Safety and security (C9, β = 0.036, t = 0.601, $p$ = 0.55) is reported to have no significant influence on the sustainability of ECs in the wake of a pandemic across the groups, but found significant within the groups of students residing in urban area with positive influence, students enrolled in computer-allied majors with positive influence and students enrolled in non-computer allied majors with negative influence. During pandemics, the only evident approach to continuing education further is to implement preventative safety and security measures for students' general well-being [395].

Urban students (β = 0.21, $t$ = 2.14, $p$ = 0.03) are fully aware of the need of preserving their health and safety during a pandemic since the virus is likely to strike the city first accompanying dense population. On this basis, students from metropolitan hometowns have accepted greater EC sustainability under COVID-19 because to their inherent demand for safety, security, and clean settings. According to the results of [384] urban residents are most likely to be sustained by safety and security, as noted by this study.

Enhancing safety and security measures for the students belonging to computer allied majors (β = 0.17, $t$ = 2.28, $p$<0.05) considerably boosts sustainability during the COVID-19 pandemic, since the pedagogy associated with the major looked to be safe and secure for them during the pandemic. While, students enrolled in non-computer allied majors (β = -0.22, $t$ = -2.30, $p$<0.05) perceived relatively less than expected on the sustainability with increasing safety and security measures, possibly due to a fear of academic loss [277] due to restricting accessibility by over-imposing safety and security.

## Effect of curriculum delivery

Curriculum delivery (C10, β = 0.141, t = 2.073, $p$ = 0.039) has established significant contribution to the long-term sustainability of ECs in the context of a pandemic across diversity. While education transformation occurs through the present of social, cognitive, and instructions, students' traits such as SES status and geographic are unaffected by inferring sustainability in terms of curriculum delivery. This is possibly why the study signified no impact of curriculum delivery on sustainability of ECs across these groups, making it similar to the research of [106]. However, it has made significant impact within the groups including male, high scorers, students enrolled in non-computer allied majors, students opting hybrid mode, students preferring non-interactive-offline information source and community.

Male students (C10, β = 0.141, $t$ = 2.073, $p$ = 0.039) and high academic performers (β = 0.35, $t$ = 3.29, $p$ = 0.000), who felt that stronger curriculum delivery is responsible for their perceptions of greater sustainability in the COVID-19 situation. Before the pandemic, there were many different opinions on whether or not online curriculum delivery was enough. For this reason, these groups might worry about the suitable delivery of the curriculum to sustain their academic traits and abilities. Thus, the fact that male and high academic performing students believed ECs to have more sustainability during pandemic conditions with more enhanced and refined curriculum delivery.

Furthermore, if the curriculum for non-computer associated majors, which includes of more numerical and practical topics, is not appropriately given during the pandemic,

academic loss and isolation may follow. Higher workloads and skill limitations in these majors may be problematic during the epidemic [290]. The hybrid distribution modality has demonstrated promising results for students who have returned to school following the closure of ECs during the pandemic [105,107,396]. This provides a plausible basis for the students enrolled non-computer allied majors (β = 0.25, $t$ = 2.09, $p$ = 0.04) and students opting for hybrid mode (β = 0.40, $t$ = 3.03, $p<0.05$) to recognise that higher sustainability under COVID-19 is related to increase and sophisticated curriculum delivery. In this regard, the findings of this study are comparable with those of [397], who discovered that hybrid delivery is the most effective and sustainable technique in the Indian setting during the COVID-19 pandemic.

Students gathering information from Interactive-offline (β = 0.49, $t$ = 2.79, $p$ = 0.01) like face-to-face counselling sessions, campus visits, and students influenced by community (β = 0.44, $t$ = 3.10, $p$ = 0.000) comprising friends/peers and relatives perceived that better curriculum delivery moves sustainability of ECs in a positive direction under COVID-19 pandemic. As eyewitnesses and role models are the true communicators when it comes to providing factual information, interactive-offline information and community influence seemed like to govern the estimate of sustainability of ECs by the means of curriculum delivery under the pandemic.

Students who gathered information from interactive-offline (β = 0.49, $t$ = 2.79, $p$ = 0.01) sources such as face-to-face counselling sessions and campus visits, and students influenced by community (β = 0.44, $t$ = 3.10, $p$ = 0.000) such as friends/peers and relatives perceived that better curriculum delivery moves sustainability in a positive direction under the COVID-19 pandemic. Accurate information about curriculum delivery had been provided by eyewitnesses and role models who are the substantial communicators. For this reason, students utilizing interactive-offline information sources and community impact have seemed to perceive greater estimation of the sustainability of ECs with the accurate curriculum delivery during the pandemic.

## Effect of value for money

Cost-effectiveness, time, and effort expended are particularly important in pandemic scenarios since they relate to mental and health issues. However, the associated trait with ECs, value for money (C11, β = 0.081, t = 1.638, $p$ = 0.102) was found insignificant in backing sustainability of ECs in the context of a pandemic. However within and across the groups includes; native places (rural students), pre-college performance (low scorers), major choice (students enrolled in computer allied majors), curriculum delivery mode (students opting for online and hybrid), information sources (students who utilized non-interactive-online and interactive-online sources), and human influence (ECs representatives), it is has made impact on sustainability.

In this study, rural students (β = 0.14, $t$ = 2.03, $p$ = 0.04), lower academic performers (β = 0.14, $t$ = 2.18, $p$ = 0.03), students choosing online pedagogy (β = 0.29, $t$ = 2.75, $p$ = 0.05) and students enrolled in computer-allied majors (β = 0.21, $t$ = 3.20, $p$ = 0.000) had positive impressions on sustainability of ECs by the virtue of value of money. For the rural students, this kind of investment leads to substantial benefits such as quality of life, income, and social prestige and culture. Value for money refers to awarding benefits in terms of ability, quality, income, and social prestige in contrast to time and effort investments made by parents and students. Computer allied majors are associated with high earning and social prestige. If the expected return is larger than expected, students may be prepared to invest more money or time [194]. Online education is extremely cost-effective in terms of convenience and advantages for the students preferring it [104,398]. As a result, these students perhaps had perceived higher sustainability due to better value for money. Nonreactive and offline sources (β = 0.21, t = 2.62,

$p$ = 0.01) like website, interactive-online sources (β = 0.22, t = 2.11, $p$ = 0.04) like social media and ECs representatives (β = 0.19, t = 2.59, $p$ = 0.01) had positive impact on sustainability which are the effective ways of communicating the advantages of a career in engineering and proposing the best EC option among the available [374] may be the probable reasons for enhancing perceptions of greater sustainability with greater value for money benefits.

However, in the case of students opting for hybrid pedagogy (β = -0.27, $t$ = -2.16, $p$ = 0.03), the influence of value for money was relatively less than expected in assessing sustainability of ECs. Hybrid delivery might result in doubling students' time, efforts, and cost, and therefore is not desirable during the pandemic situations. This may be the cause of the students' perception preferring hybrid mode that as value for money increases, sustainability of ECs under COVID-19 falloffs.

Nevertheless, students who favored a hybrid pedagogy (β = -0.27, $t$ = -2.16, $p$ = 0.03) acknowledged value for money, relatively smaller than predicted in determining the sustainability of ECs. Hybrid delivery is unfavorable in pandemic since it might double the time, effort, and expenses creating the unstable and panic situation for the students and parents. This might be the reason why students preferring hybrid mode believed relatively lesser sustainability than anticipated with the perceptions on value for money rises.

In regard to RQ3, the statistical findings showed that across the students' traits, including their demographic, socioeconomic, geographic, and academic performance had no disparities on the bond- sustainability of ECs formed on account of ECs' attributes. Nonetheless, diverse psychological and behavioral traits are noted with respect to their liking/choices/preferences across the groups major choice, type of curriculum delivery, access to information sources, and human influence admitted in perceiving the sustainability of ECs accumulated on account of ECs' characteristics. On the other hand, sustainability was predicated on account of EC's characteristics within each sub-group of diversity with varied strength of association.

## Practical implications, visionary suggestions, and contribution

The following managerial implications and visionary recommendations are anticipated for repositioning of ECs, endorsed for effective performance during the COVID-19 pandemic based on the study's findings.

### Repositioning

During the COVID-19 pandemic, proximity, image and reputation, quality of education and curriculum delivery are the prime governing characteristics for experiencing greater sustainability under COVID-19, based on the results of the current study.

College practices governing high-esteem, high-culture, and high-prestige are crucial for building a positive reputation and image. With effective governance and leveraging quality infrastructure and facilities along with modern, cutting-edge technology that ensures risk-management measures and sustainability under the pandemic, it is possible to reestablish the image and reputation for making positive impressions. By offering a channeled and integrated communication platform for disseminating new happenings and accomplishments, ECs can eventually activate their 'touch points' to provide positive remarks and foster massive network of relationships; both way online and offline. One action can provide a double advantage for ECs. Initially, delivering high-quality education and services will add to reputation and image. The second benefit is that it will improve students' reliability, confidence and trust in EC's commitment to provide high-quality services during pandemic. Eventually, ECs must develop co-creating mechanism to deliver and process vital information about their services and

proposals so that students and their influencers may make well-informed judgments about EC choice decisions for them.

If students were unable to fully make it to ECs when the viral severity persists, ECs should be able to reach them virtually for certain fundamental theoretical, skill enhancement certifications, and for the subjects simple to comprehend. Then bringing them in-person for practical sessions and challenging subjects in small groups while adhering to social distance standards is the only apparent solution to defend pandemic severity. Yet, such a hybrid delivery mode should sustain the ECs' commitment to develop competent engineers while rearranging the curriculum with a balanced study-load. New generation, Gen Alpha is believed to be magnetized to multi-choice curriculum that permits many multi-disciplinary fields at multi-levels and online education can be valuable in a way that permits growing choices, diversity, creativity, and connections and hence can be a useful pathway for ECs. As per the policy framework of New Education Policy, India, the notion of 'inclusive curriculum' may be operative during pandemic. If the pandemic lasts for a long time with severity, the ECs will have to look up other options like blended learning, challenge-based learning and problem based learning are quite feasible during pandemic. Last options are extremely limited and not feasible for existing ECs but promising for new establishments of ECs that include setting up new small campuses and moving to remote locations with all assets.

The physical assets of ECs are immobile, therefore there isn't much they can do to improve on proximity. Yet, this study foresaw the significance of proximity to hometown in determining a ECs' sustainability under COVID-19 pandemic. Local ECs are therefore enforced to provide top-notch education while adhering to COVID-19 protocols in order to seize this breakthrough to avoid local brain drains. Although ECs located far from students' hometown can still entice diverse enrollments during pandemic by offering reputable and emerging majors, enhancing brand reputation and providing appropriate delivery as the situation permits that sustain students' interest active and preventing academic loss. ECs in remote and rural areas will benefit from being able to provide courses on-site, however, must be resilient to mitigation management and employ successful promotional techniques to foster co-creation of trust towards ECs.

## Inclusion of diversity

Although the situational notion of "sustainability under pandemic" is anticipated to have no influence on the students' characteristics including demographic, socioeconomic, geographic, and pre-college performance, there is necessity to analyze the diverse students on their psychological and behavioral traits at the micro level to make EE sustainable, ensuring their inclusivity into the ECs. Realization of "all are not one", is important for the ECs under the pandemic. Considering the possibility of future pandemics, ECs bear a larger obligation to create learning opportunities for EE that take account of future diversity, inclusivity and equity. The results of the current study's regression analysis of students' and ECs characteristics during pandemic conditions may be helpful in this regard.

For instance, by providing quality based technology and services, appropriate teaching, and impressing reputation on the minds, male students can be encouraged to enroll on campus. For underrepresented groups, like as female and lower SES students, a team of faculty members acting as counsellors or mentors can be useful in fostering interest and resilience that inspire a sense of belonging, enabling their college choice during the pandemic. Prejudices towards based on stereotypes such as 'muscularity', 'richer's game', 'superior culturist', 'chilly climate', 'technical', and 'tough and hard' should be alleviated for the successful journey—inclusion to empowering uninterrupted. In reviewing inequalities in engineering, it is

important to understand and respond to; diversity that ensuring all are highlighted, inclusion that enables sense of belonging and confidence in all, equity that acknowledges satisfaction among all. That is why the role of psychological trainers and sustainability professionals in ECs are indispensable in guarding equity and developing sustainability of ECs that enables competent engineers with mental well-being. Next, to include and retain diversity, socialization through engineering culture and values should be commenced with the start of first academic session and not after graduation.

Higher SES and high performing students are highly impacted by the image and reputation of ECs, therefore ECs' participation in government's numerous efforts, including COVID-19 preventive activities, may be eye catching in the media adding to their social standing. High-quality amenities with greater safety and security, particularly those connected to health, and hygienic dorms appropriate during pandemics, should be made available to urban students from high-culture backgrounds. Since remote and rural-based industries and enterprises are currently the safest and most secure places during pandemic to pursue a career for urbanized and computer-related enrollments, campus placement offers of ECs should be expanded to include these sectors.

According to the study's findings, onsite education was proposed by students who felt it to be suitable despite the pandemic condition. This is true because education is more than merely imparting knowledge; it also involves the human touch, facial reading, and encouragement of social, affective, and epistemic relationships among students. Further if EE has to be ethical and competent in enhancing skills, field and lab experience is required which is feasible only with physical existence of students and equipment together. High performing students and students with non-computer allied majors, must be then exposed to adequate hybrid or onsite delivery with pandemic measures, depending on the severity of the pandemic crisis, to sustain their morale and interest. Delivery of the curriculum to rural and low-scoring students should occur in a way that avoids adding more time, money, effort and study load. The expense of digital devices and the associated technology should not result in additional costs, mental stress and trauma due to technology handling while delivering the curriculum. In any case shift must be towards professional and social aspects of engineering rather than technical.

ECs should in house all required measures that meet diverse expectations of 'fit for the purpose' under the COVID-19 pandemic. How quickly and how far ECs acquire this, will determine if they can build a "house of resilience and reliance" to withstand a pandemic. By raising awareness reduces gap for deprived aspirants. Here, the importance of human influence and promotional information sources are crucial to positioning ECs as a COVID-19-compliant organization. For promoting "word-of-mouth" and making its "sustainability" viral, the role of ECs' in built stakeholders such as faculty, current, and alumni students is crucial. ECs should motivate their stakeholders to follow social media pages, communicate important information, and participate in snowballing the social reactions creating a future market for ECs during a pandemic crisis.

## Research contribution

To the best of our knowledge, this study is the first to provide insights into the performance of choice characteristics to evaluate the selection of ECs during the COVID-19 pandemic. It is also believed to be the first to develop the concept of a new gadget—"sustainability" relating to new enrollments, which is recognized as a crucial tool for assessing choice characteristics in pandemic situations. The study has successfully investigated the connections between students' characteristics and ECs' characteristics along with its sustainability during the pandemics, which is the primary contribution of this study. These characteristics represent how students

view their attitudes, personalities, values, learning perceptions, motivation, lifestyles, and preferred communication methods from an intellectual and behavioral perspective, all of which are in response to ECs sustainability under pandemic leading college selection decisions.

Regression modelling was employed in this study to represent quantitative and statistical results that may be used as a yardstick for the effective performance of EC's in a competitive market. After learning the insightful analysis on the effectiveness of choice characteristics to regulate the sustainability of ECs under the COVID-19 framework, academicians and policy makers will have a substantial hope and opportunity of attracting diverse enrollments and reestablishing normalcy. The study might make it easier for future studies and administrators on developing suitable policies and best practices.

The pandemic influence had all the makings of a losing cause for new enrollments. For academics, it was typically mysterious homework, and for marketing professionals, it turned into time-consuming guesswork. This research indirectly responded itself to finding answers to three appealing C's: continuity, communication, and community, all of which have the potential to draw in and retain a larger number of students. While this study, firmly confirms the observable radical tendency of the pandemic, it provides practicable approach for ECs, even if the pandemic lasts for a very long time or coexists with us with high intensity. It is imperative to improve the sustainability of private institutions as the bulk of the lakhs of future engineers of the country will come from private-aided ECs. The findings of this research have promising potential in featuring flexible routes for ECs and enabling the vision of New Education Policy [399] of achieving a 50% gross enrolment ratio by 2035. Last, the study with adding new, crucial and substantial information concerning the new term—sustainability of ECs during pandemics, nevertheless, it has greatly validated existing literature claiming students psychological and behavioral change during pandemic.

## Conclusion

Worldwide higher education has been impacted by COVID-19, but particularly pulling Indian ECs' padlocked doors down. Students and ECs both were at more at risk while doing nothing during the start of pandemic. Prospective students grow in order to fulfil their longtime desire of attending college. ECs perform to serve for the benefit of students, industry society and the whole nation at large. The attitude of both, potential students and ECs should be tuned to the "show must go on" principle. As engineers are the driving force behind progress at every level, COVID-19 should be seen as a catalyst moving ahead. Students look into several ECs to find those to which they can invest the most time, effort, and resources for the highest likelihood of acquisition their intended goals. Nonetheless, ECs are looking for students with a wide range of interests who will contribute to their vibrant campus. Analytical mapping of choice characteristics of the current work has provided an educational vaccine for ECs that may create antibodies to increase future enrollments and smoothen the sensation of "college choice" for aspiring students. Sustainability materialized in this study must be seen as an expanded opportunity and not as a binding constraints.

In Feb of 2021, when the COVID-19 pandemic had already been going strong in India since a year, we began our study survey. Students who were enrolled to first year of engineering for the 2020–2021 academic year were among the first in India to experience such a challenging progression of 'college choice' after the pandemic's introduction in India. Within this sphere the study moved into the investigation of choice characteristics that influenced students' decision about ECs' selection.

The primary objective of this study was to explore diverse students' perceptions of choice characteristics related to their traits and ECs influencing college choice decisions under the

COVID-19 pandemic, and to uncover any possible relationships between them, has been successfully accomplished. Three research questions were qualitatively examined with the statistical confirmation of related hypothesizes. First, the study discovered that, along with the development of an entirely novel concept called "sustainability under COVID-19", the traditional choice characteristics associated with the ECs that are usually prevalent in decision making during pre-pandemic period, are still applicable even during the pandemic. Second, the optimum design of ECs' sustainability in a global pandemic is determined by four traditional characteristics of ECs; proximity, image and reputation, educational quality, and curriculum delivery. Next, the findings showed that ECs have several characteristics that render them sustainable under the pandemic within and across (in between the groups) student heterogeneity due to psychological and behavioral biases on liking, choices and preferences, providing a framework on which ECs could strengthen themselves on repositioning to capture potentially diverse enrollments. Yet, this induced impact is consistent within (not across) demographic, geographic, socioeconomic, and academic performance traits of newly enrolled students.

Lastly, we emphasized our visionary suggestions and practical implications, focusing on taking collective actions to promote EE and make informed decisions about the ECs' selection. During the pandemic, ECs may see improvements in benchmarked enrolment levels if they use some truly encouraging student-centric strategies meeting students' changing needs as discussed herein. Even if we have to coexist with COVID-19 forever or more pandemics, research contribution of this study may see ECs (existing as well forthcoming) to undergo a fundamental transformation.

## Limitations and future research

Though the study survey was conducted during the COVID-19 pandemic, the regression model of choice characteristics made it appropriate to consider even in non-pandemic settings. Future research that takes into account a large population, region, or country is encouraged with the full pandemic constraints open. Further though we have limited our analysis to understand the perceptions of students as a prime customers and future studies considering other stakeholders, such as faculty and alumni is anticipated.

When the research survey was conducted, the pandemic in India had some constraints, therefore as per our convenience, we narrowed the study sample purposefully to the North Maharashtra region of India. Nonetheless, we believe that the findings may be applied to other regions, given that pandemics have had similar effects all around the world. The current study's sincere approach and authenticity in examining multidimensional relationships of newly developed gadget- sustainability and traditional choice characteristics related to ECs across diverse students has however made it appropriate to take it into consideration for other regions or even globally. Realizing the benefits, plenty of avenues for research, are now attainable for other higher educational discipline studies like management, agriculture, or health-related courses.

In its first attempt, even though this study has shed light on considerable advancement in detailing sustainability as a new choice characteristic, there are numerous opportunity for future research to create a scale that is more precise and in-depth. Future study that includes numerous choice characteristics such as campus life, institutional brand etc. and students' other psychological and behavioral traits that changes as the situation is anticipated. Similar study can be conducted longitudinally to judge further effectiveness of the ECs' sustainability in pandemic situations.

## Author Contributions

**Conceptualization:** Prashant Mahajan, Vaishali Patil.

**Data curation:** Prashant Mahajan, Vaishali Patil.

**Formal analysis:** Prashant Mahajan, Vaishali Patil.

**Investigation:** Prashant Mahajan, Vaishali Patil.

**Methodology:** Prashant Mahajan, Vaishali Patil.

**Project administration:** Prashant Mahajan, Vaishali Patil.

**Resources:** Prashant Mahajan.

**Software:** Prashant Mahajan.

**Supervision:** Prashant Mahajan, Vaishali Patil.

**Validation:** Prashant Mahajan, Vaishali Patil.

**Visualization:** Prashant Mahajan, Vaishali Patil.

**Writing – original draft:** Prashant Mahajan, Vaishali Patil.

**Writing – review & editing:** Prashant Mahajan, Vaishali Patil.

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
