## [Decision Letter · Decision Letter 0]

7 Jun 2023

PONE-D-23-08845

‘College Choice’ under the COVID-19 pandemic: Sustainability of Engineering Campuses for future enrollments.

PLOS ONE

Dear Dr. Mahajan,

Thank you for submitting your manuscript to PLOS ONE. After careful consideration, we feel that it has merit but does not fully meet PLOS ONE’s publication criteria as it currently stands. Therefore, we invite you to submit a revised version of the manuscript that addresses the points raised during the review process.

We look forward to receiving your revised manuscript.

Kind regards,

JOANNA ROSAK-SZYROCKA, Assistant Professor

Academic Editor

PLOS ONE

Journal Requirements:

3. Please amend your current ethics statement to address the following concerns:

a) Did participants provide their written or verbal informed consent to participate in this study?

   "No. The funders had no role in study design, data collection and analysis, decision to publish, or preparation of the manuscript."

Reviewers' comments:

Reviewer's Responses to Questions

**Comments to the Author**

1. Is the manuscript technically sound, and do the data support the conclusions?

Reviewer #1: Yes

Reviewer #2: Yes

2. Has the statistical analysis been performed appropriately and rigorously? 

Reviewer #1: Yes

Reviewer #2: Yes

3. Have the authors made all data underlying the findings in their manuscript fully available?

Reviewer #1: Yes

Reviewer #2: Yes

4. Is the manuscript presented in an intelligible fashion and written in standard English?

Reviewer #1: Yes

Reviewer #2: Yes

5. Review Comments to the Author

Reviewer #1: In their research, the authors are focusing on a less-researched topic of how the preferences of prospective students changed after the covid. The research is really gap-filling and provides a comprehensive overview and dynamics of the engineering higher education in India and internationally too.

The title covers and reflects the content, the abstract is compact and clear, the keywords are relevant.

The introduction sets the topic and the research into context, presents the motivation and the research goals as well.

The literature review is not just going about processing the relevant sources but provides a theoretical framework too, in a very detailed and structured way.

The methodlogy is described and introduces very detailed too. The authors selected the appropriate methodological toolset for executing this research, and applied it well.

The results are supported by the methods and provides clears and well understandable information and explanations.

Conclusions are based on the results, implications and limitations are presented as well.

Reviewer #2: I think the paper is very interesting and it’s contribute to the new knowledge in the field. It’s good prepared based on the extensive literature analysis and use various statistical methods to data analysis.

There are following problems worth improving in the paper:

• In the theory please describe more about relations between sustainability and COVID pandemic. How the pandemic have impacted various problems connected with sustainability.

• In the methodology describe in brief statistical methods used in the paper.

6. PLOS authors have the option to publish the peer review history of their article (what does this mean?). If published, this will include your full peer review and any attached files.

Reviewer #1: No

Reviewer #2: No

<quillbot-extension-portal></quillbot-extension-portal><quillbot-extension-portal></quillbot-extension-portal>

---

## [Author Response · Author response to Decision Letter 0]

17 Jun 2023

Respected Reviewers

First and foremost, thank you for reviewing our research paper. Your inspiring remarks have enlightened us. Further, we are grateful to you for offering helpful suggestions and recommendations that helped make this updated paper more constructive.

Reviewer#1: No specific requirement for revision

Reviewer#2: Following matter / text added to improvise the manuscript. 

Comment (1)

“In the theory, please describe more about relations between sustainability and COVID pandemic. How has the pandemic impacted various problems connected with sustainability?”

Response by authors

As per reviewer’s suggestion following text is added. 

“Sustainability in education is all about designing practices that can be scaled or right-sized without depleting resources or exclusion of groups (301). Sustainability became more important in higher education right before the outbreak of the pandemic, with HEIs developing customized medium- and long-term strategy plans focusing on sustainability. However, the COVID-19 pandemic has halted several initiatives and strategic plans. Most HEIs were compelled to adjust their focus from 'sustainability' to 'survival' as a result of the pandemic influence, blocking the path of transformative change. If sustainability is to be revived as a development objective, then during pandemics, the prerequisite condition for HEIs is the development of effective plans and policies that attract new enrolments, fill seats to capacity (full resources), and raise revenues, without which sustainable system may be hindered to obtain desire effects (302). As sustainability relates to the capability of being supported or maintained or kept going (303), it must be reviewed during pandemic.”

Please refer line numbers 620-631 in the revised manuscript highlighted in RED TEXT.

Comment (2)

In the methodology describe in brief statistical methods used in the paper.

Response by authors

As per reviewer’s suggestion following text is added 

“To validate hypothesis H1, Analysis of Variance (ANOVA) was utilized to identify the significant difference in students’ perceptions of sustainability of ECs under the COVID-19 pandemic. Hypothesis H2 was tested by Regression Analysis to detect the accountable relationship in between the sustainability of ECs under COVID-19 pandemic and the ECs’ characteristics. ANOVA was performed to predict the statistical significant across the students’ characteristics to explain regression model representing the relationship in between the sustainability of ECs under COVID-19 pandemic and the ECs’ characteristics to justify Hypothesis H3. All above statistical analysis were performed by Statistical Product and Service Solutions (SPSS).”

Please refer line numbers 787-795 in the revised manuscript highlighted in RED TEXT.

Once again thank you!

AUTHORS

---

## [Decision Letter · Decision Letter 1]

7 Sep 2023

‘College Choice’ under the COVID-19 pandemic: Sustainability of Engineering Campuses for future enrollments.

PONE-D-23-08845R1

Dear Dr. Prashant Mahajan,

We’re pleased to inform you that your manuscript has been judged scientifically suitable for publication and will be formally accepted for publication once it meets all outstanding technical requirements.

Kind regards,

JOANNA ROSAK-SZYROCKA, Assistant Professor

Academic Editor

PLOS ONE

Additional Editor Comments (optional):

Reviewers' comments:

<quillbot-extension-portal></quillbot-extension-portal><quillbot-extension-portal></quillbot-extension-portal>

---

## [Editor Report · Acceptance letter]

15 Sep 2023

PONE-D-23-08845R1 

‘College Choice’ under the COVID-19 pandemic: Sustainability of Engineering Campuses for future enrollments. 

Dear Dr. Mahajan:

I'm pleased to inform you that your manuscript has been deemed suitable for publication in PLOS ONE. Congratulations! Your manuscript is now with our production department. 

Kind regards, 

on behalf of

Dr. JOANNA ROSAK-SZYROCKA 

Academic Editor

PLOS ONE